# Historically inconsistent productivity and respiration fluxes in the global terrestrial carbon cycle

Jinshi Jian [1,2,3,4✉], Vanessa Bailey [5], Kalyn Dorheim [2], Alexandra G. Konings [6], Dalei Hao [7], Alexey N. Shiklomanov[8], Abigail Snyder [2], Meredith Steele[9], Munemasa Teramoto[10,12], Rodrigo Vargas [11] & Ben Bond-Lamberty [2]

The terrestrial carbon cycle is a major source of uncertainty in climate projections. Its dominant fluxes, gross primary productivity (GPP), and respiration (in particular soil respiration, $R_S$), are typically estimated from independent satellite-driven models and upscaled in situ measurements, respectively. We combine carbon-cycle flux estimates and partitioning coefficients to show that historical estimates of global GPP and $R_S$ are irreconcilable. When we estimate GPP based on $R_S$ measurements and some assumptions about $R_S$:GPP ratios, we found the resulted global GPP values (bootstrap mean $149^{+29}_{-23}$ Pg C yr$^{-1}$) are significantly higher than most GPP estimates reported in the literature ($113^{+18}_{-18}$ Pg C yr$^{-1}$). Similarly, historical GPP estimates imply a soil respiration flux ($Rs_{GPP}$, bootstrap mean of $68^{+10}_{-8}$ Pg C yr$^{-1}$) statistically inconsistent with most published $R_S$ values ($87^{+9}_{-8}$ Pg C yr$^{-1}$), although recent, higher, GPP estimates are narrowing this gap. Furthermore, global $R_S$:GPP ratios are inconsistent with spatial averages of this ratio calculated from individual sites as well as CMIP6 model results. This discrepancy has implications for our understanding of carbon turnover times and the terrestrial sensitivity to climate change. Future efforts should reconcile the discrepancies associated with calculations for GPP and Rs to improve estimates of the global carbon budget.

[1] State Key Laboratory of Soil Erosion and Dryland Farming on the Loess Plateau, Northwest A&F University, Yangling 712100, China. [2] Pacific Northwest National Laboratory, Joint Global Change Research Institute at the University of Maryland–College Park, 5825 University Research Court, Suite 3500, College Park, MD 20740, USA. [3] University of Chinese Academy of Sciences, Beijing 100049, China. [4] Institute of Soil and Water Conservation, Northwest A & F University, Yangling, Shaanxi 712100, China. [5] Biological Sciences Division, Pacific Northwest National Laboratory, Richland, WA 99354, USA. [6] Department of Earth System Science, Stanford University, 473 Via Ortega, Room 140, Stanford, CA 94305, USA. [7] Atmospheric Sciences and Global Change Division, Pacific Northwest National Laboratory, Richland, WA 99354, USA. [8] NASA Goddard Space Flight Center, 8800 Greenbelt Rd., Building 33, Greenbelt, MD 20771, USA. [9] School of Plant and Environmental Sciences, Virginia Tech, 183 Aq Quad Ln, Blacksburg, VA 24061, USA. [10] National Institute for Environmental Studies, 16-2 Onogawa, Tsukuba 305-8506, Japan. [11] Department of Plant and Soil Sciences, University of Delaware, Newark, DE 19716, USA. [12] Present address: Arid Land Research Center, Tottori University, 1390 Hamasaka, Tottori 680-0001, Japan. ✉email: jinshi@vt.edu

The terrestrial carbon sink removes about a quarter of anthropogenic $CO_2$ emissions[1] but is highly variable in time and space depending on climate. The magnitude of gross primary productivity (GPP) is therefore one of the largest sources of uncertainty in predicting future trajectories of global temperature[2]. For example, GPP is a first-order control on plant turnover times, a dominant uncertainty term in the terrestrial carbon sink[3]. There has been substantial progress in quantifying and constraining GPP and other major global carbon fluxes, typically using models driven by satellite remote sensing[4–7] and upscaled in situ ecosystem-scale flux measurements[8,9]. Recent syntheses[7,10] suggest that global GPP is 120–125 Pg C yr$^{-1}$, and such estimates from the literature (GPP$_{lit}$) have been incorporated into synthesis efforts such as the Global Carbon Project[1] as well as model benchmarking frameworks[11]. The magnitude of terrestrial GPP thus has implications for the dynamics and resilience of the terrestrial C sink in the face of global environmental change[12,13].

Global GPP is roughly balanced by ecosystem-to-atmosphere respiratory fluxes. The difference between these two major fluxes, minus smaller fluxes such as fire and lateral (e.g., dissolved, particulate) organic carbon losses, comprises the terrestrial C balance[1]. Terrestrial ecosystem respiration is dominated by the soil-to-atmosphere $CO_2$ flux (soil respiration or R$_S$), the combined flux generated by microbial and plant root respiration. Respiration is rarely estimated, even indirectly, from satellite observations, and thus global R$_S$ is generally derived by upscaling in situ measurements[14–16]. Published R$_S$ estimates from the literature (Rs$_{lit}$) range from 68 to 109 Pg C yr$^{-1}$ (Supplementary Table 1), with a central range of 85–90 Pg C yr$^{-1}$ [17]. Because GPP and R$_S$ are physiologically linked, the biophysical balance between GPP and R$_S$ could be used as a constraint on the global carbon budget. To date, however, no attempt has been made to quantify how consistent these independent GPP and R$_S$ estimates are at the global scale. This study compares these two large carbon fluxes and the results emphasize the importance of cross-comparing datasets and models to understand terrestrial carbon cycling as well as future climate change.

## Results and discussion

**Inconsistency between photosynthesis and soil respiration.** We partitioned global Rs$_{lit}$ estimates into microbial and root respiration based on all available (published) partitioning values, and calculated distributions of the resulting implied GPP (GPP$_{Rs}$) using literature estimates of net primary production (NPP) and root-to-shoot respiration ratios (Supplementary Figs. 1–7). Using a nonparametric bootstrap, we generated 10,000 such GPP$_{Rs}$ estimates based on random draws from Rs$_{lit}$, NPP, the partitioning parameters (see Methods and Supplementary Figs. 5, 8–10 and Supplementary Tables 1, 2), and the corresponding uncertainties. The resulting GPP$_{Rs}$ distribution was $149^{+29}_{-23}$ Pg C yr$^{-1}$ (mean ± 95% confidence interval; Fig. 1), which contrasts with the GPP$_{lit}$ average of $113^{+18}_{-18}$ Pg C yr$^{-1}$. The intersection of these two distributions is 127.6 Pg C yr$^{-1}$ (Fig. 1), a point at the 95.2% quantile of GPP$_{lit}$ and the 9.8% quantile of GPP$_{Rs}$. The null hypothesis (that these distributions are from the same underlying population) is highly unlikely: $t_{49} = -12.68$; $P < 0.001$. What characterizes the small number of estimates consistent with both GPP$_{lit}$ and GPP$_{Rs}$? Bootstrap draws in the overlap region were characterized by low root contribution to R$_S$ (averaging 34% below the intersection point, versus 42% above it) and high root contribution to autotrophic respiration (45 vs. 38%, respectively; Supplementary Fig. 11), resulting in low GPP$_{Rs}$ values.

We performed a comparative analysis of published data to derive R$_S$ from GPP, partitioning GPP$_{lit}$ into NPP and belowground autotrophic respiration components, while accounting for other carbon loss pathways (see Methods). The resulting implied Rs$_{GPP}$ (i.e., the global R$_S$ as implied by GPP$_{lit}$, $68^{+10}_{-8}$ Pg C yr$^{-1}$; Fig. 1) is highly unlikely to be consistent with Rs$_{lit}$ values ($87^{+9}_{-8}$ Pg C yr$^{-1}$; see Methods). Only 1.8% of the Rs$_{lit}$ distribution in Fig. 1 is below the intersection point of 78.2 Pg C yr$^{-1}$, and only 2.5% of the Rs$_{GPP}$ distribution is above it. This is strong evidence against the null hypothesis that these curves are mutually consistent (i.e., that they represent the same underlying population, $t_{23} = -11.59$; $P < 0.001$). The overlap between these distributions is characterized by high GPP$_{lit}$ (averaging 125.6 Pg C yr$^{-1}$, versus 112.5 Pg C yr$^{-1}$ below the intersection point), high NPP, and a high contribution of roots to overall autotrophic respiration (46 and 39% for above and below the intersection point, respectively; supplementary Fig. 12). The cumulative result of these values produced the small percentage of Rs$_{GPP}$ draws consistent with Rs$_{lit}$.

We identified sources of variability in Fig. 1 using a variance decomposition procedure to explore which parameters were both uncertain and influential in the distribution of GPP$_{Rs}$ and Rs$_{GPP}$ (Table 1). Variability in GPP$_{Rs}$ was dominated (63% of total variance) by uncertainties in the ratio of root respiration to total autotrophic respiration, for which field measurements are limited. Other influential variables were variance in global Rs$_{lit}$ (12%) and the root contribution to total R$_S$ of a desert, wetland, and savanna (other, 7%). For bootstrapped Rs$_{GPP}$, uncertainty in GPP$_{lit}$ was the largest (35%) contributor to variability, with root contribution to total R$_A$ of cropland, savanna, grassland, and wetland (other, 32%) and global NPP (28%) also large. No other factor contributed more than 2% for variability in GPP$_{Rs}$.

We also employed a second, complementary approach, one independent of any assumptions about carbon partitioning. In this step, we compared site-level measurements of R$_S$ and GPP from a global soil respiration database (SRDB[18]) and FLUXNET[19]. These were compared against the same global GPP$_{lit}$ and Rs$_{lit}$ estimates shown in Fig. 1. The site-level R$_S$:GPP ratios (i.e., the values directly reported by investigators and compiled in SRDB) averaged 0.56 ± 0.26 (Fig. 2), very similar to the R$_S$:GPP ratios from combining SRDB and FLUXNET data (0.54 ± 0.85). These were both significantly ($P < 0.001$ based on a nonparametric Wilcoxon test) lower than the Rs$_{lit}$:GPP$_{lit}$ ratios of 0.72 ± 0.11.

We found no evidence that this difference was driven by a lack of spatial representativeness in the global distribution of SRDB data. For example, the arithmetic mean of the R$_S$:GPP ratio in the SRDB is 0.56, and 0.57 when weighted by vegetation areas globally. We highlight that this does not mean that the difference cannot be influenced by sampling errors related to the sparsity of the underlying measurements. Figure 2 also shows R$_S$:GPP and R$_H$:GPP values from models in the Coupled Model Intercomparison Project phase 6 (CMIP6)[20] at both local (grid cell site-level) and global scales. These models are global in extent, similar to satellite data products, but their explicit physiological processes mean that their R$_S$ outputs are constrained by GPP. In the CMIP6 models examined, R$_S$:GPP values were 0.609 ± 0.11 at both the global scale (i.e., the ratio of the models' global fluxes) and the scale of individual grid cell site-level, which were significantly lower (W = 375,206, $P < 0.001$) than global Rs$_{lit}$:GPP$_{lit}$ values shown in Fig. 2.

The R$_H$:GPP ratios from CMIP6 models do not significantly differ from the global R$_H$:GPP ratio from the literature ($P = 0.93$, Fig. 2d), indicating that the low R$_S$:GPP ratio of the CMIP6 models (Fig. 2b) is likely due to too-low R$_{root}$ values, either because the fluxes are incorrectly parameterized, or because the allocation of carbon across different pools is incorrectly represented. Carbon allocation is a notable weak link in current

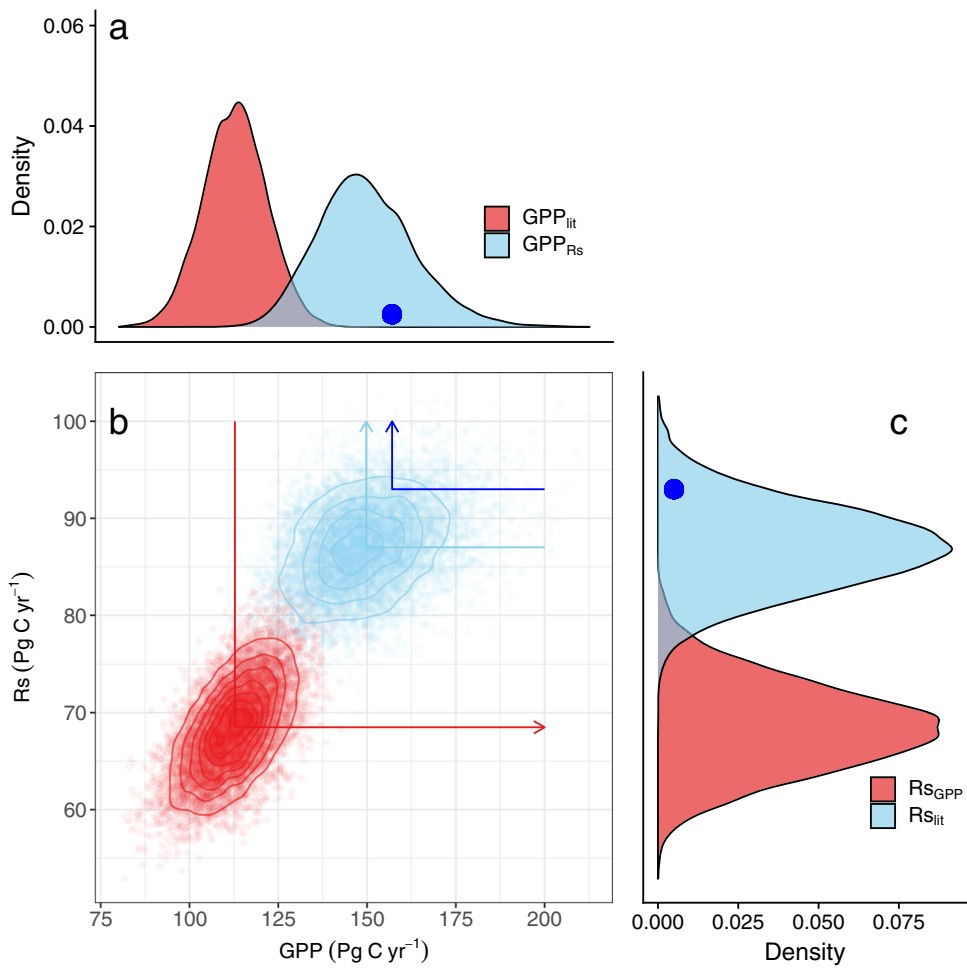

**Fig. 1 Distribution and comparison of annual global soil respiration ($R_S$) and gross primary productivity (GPP). a** Distributions of global gross primary productivity ($GPP_{lit}$ and $GPP_{Rs}$); **b** Joint distribution of annual global soil respiration ($R_S$) and gross primary productivity (GPP); **c** Distribution of global soil respiration ($Rs_{lit}$ and $Rs_{GPP}$) estimates. Two distributions are shown: literature-reported GPP ($GPP_{lit}$) versus GPP implied by those $R_S$ estimates ($GPP_{Rs}$); or literature-reported $R_S$ ($Rs_{lit}$) versus $R_S$ implied by those GPP estimates ($Rs_{GPP}$); Distributions are based on 10,000 random draws of the underlying estimates from published literature (summarized in supplementary Fig. 8). The red arrow represents from $GPP_{lit}$ to calculate $Rs_{GPP}$, the light-blue arrow represents from $Rs_{lit}$ to calculate $GPP_{Rs}$, and the blue dots and line represent $R_S$ from the random forest model developed in this study and based on that to calculate the $GPP_{Rs}$. The arrows and direction corresponding to the arrows in supplementary Fig. 1.

ESMs due to both a lack of empirical observations and uncertainty over the underlying physiological mechanisms, and the $R_S$:GPP ratio could be a valuable model benchmark to constrain root allocation. An even stronger approach, in our view, is to use data assimilation in model benchmarking efforts[11] to estimate multiple C and biogeochemical fluxes simultaneously, so that they are constrained by each other.

These independent lines of the analysis demonstrate that $GPP_{lit}$ and $Rs_{lit}$, the historical global flux estimates reported in the published scientific literature, are almost certainly inconsistent with each other. One possible interpretation of this problem is that many published global GPP estimates are biased low. If the mean of the $GPP_{Rs}$ distribution (149 Pg C yr$^{-1}$) in Fig. 1 is the actual global flux, for example, that would be close to that implied by atmospheric $^{18}O$:$^{16}O$ ratios of $CO_2$, which suggest that a global GPP of 150–175 Pg C yr$^{-1}$ is needed to explain rapid $CO_2$ cycling times[21]. A similar conclusion was reached in recent studies using novel methods such as $O_2$:$CO_2$ ratios associated with the land carbon exchange[22] as well as GPP derived using solar-induced fluorescence (SIF) data assimilation[5].

In an effort to derive new and independent estimates of $R_S$ and GPP, we used $R_S$ data from a recently updated global daily $R_S$

database (DGRsD) to parameterize Random Forest (RF) models for each month, and estimated global monthly $R_S$ at a spatial resolution of 0.1° (Supplementary Figs. 13, 14). Such daily data can provide more robust estimates than do annual numbers used until now to estimate global-scale $R_S$[23]. The resulting global annual $R_S$ was 93 Pg C yr$^{-1}$, with a corresponding $GPP_{Rs}$ of 157 Pg C yr$^{-1}$ (Fig. 1), close to the mean $Rs_{lit}$ (87$^{+9}_{-8}$ Pg C yr$^{-1}$) and $GPP_{Rs}$ (149$^{+29}_{-23}$ Pg C yr$^{-1}$). This also suggests that higher GPP is a possible explanation for any discrepancy between $GPP_{lit}$ and $Rs_{lit}$, but it should be noted that DGRsD is not independent of SRDB, and therefore more evidence is needed to ensure there are no systematic biases in $Rs_{lit}$.

**Possibilities to close the gap**. A number of factors might produce too-low global $GPP_{lit}$ estimates (Table 2). We found that purely remote-sensing derived GPP values, in particular from MODIS, tended to be smaller than estimates from site-level upscaling or a mixture of remote sensing and site-based measurements (Supplementary Fig. 5), consistent with recent work on the uncertainties in GPP estimation[7,12]. Note however that if $GPP_{lit}$ groups are weighted equally (i.e., aggregated into six different groups

**Table 1 Variance decomposition for the calculation of gross primary productivity (GPP) from soil respiration ($R_S$) reported in the literature (Rs$_{lit}$), and calculation of $R_S$ from literature GPP (GPP$_{lit}$).**

| Inferring GPP from $R_S$ reported in the literature (Rs$_{lit}$, Fig. 1 and Supplementary Fig. 1) | | Inferring $R_S$ from GPP reported in the literature (GPP$_{lit}$, Fig. 1 and Supplementary Fig. 1) | |
|---|---|---|---|
| Parameter | Variance (%) | Parameter | Variance (%) |
| $R_{root}$:$R_A$ (other) | 63.0 | GPP$_{lit}$ | 34.8 |
| Rs$_{lit}$ | 12.2 | $R_{root}$:$R_A$ (other) | 31.6 |
| $R_{root}$:$R_S$ (other) | 7.0 | NPP | 27.9 |
| $R_{root}$:$R_S$ (GRA) | 6.0 | $R_A$:GPP (other) | 1.8 |
| NPP | 4.0 | $C_{fire}$ | 1.5 |
| $R_{root}$:$R_A$ (GRA) | 2.6 | $R_{root}$:$R_A$ (EF) | 1.0 |
| $R_{root}$:$R_S$ (EF) | 2.0 | $R_A$:GPP (GRA) | 0.7 |
| $R_{root}$:$R_S$ (SHR) | 1.7 | $C_{sink}$ | 0.5 |
| $R_{root}$:$R_A$ (EF) | 1.3 | $C_{herbivore}$ | 0.3 |
| $R_{root}$:$R_S$ (MF) | 0.3 | DOC | 0.2 |

Columns include parameter names (parameters were fixed, one by one, to the overall mean) and percentage of total variance explained; e.g., NPP was responsible for 27.9% of the total variance when inferring $R_S$ from GPP. See Methods and Supplementary Fig. 1 for details on each computational chain. Parameters include the ratio of root respiration to total autotrophic respiration ($R_{root}$:$R_A$), net primary production (NPP), the ratio of root respiration to total soil surface respiration ($R_{root}$:$R_S$), the ratio of autotrophic respiration to GPP ($R_A$:GPP), carbon lost to fire ($C_{fire}$), carbon consumed by herbivore ($C_{herbivore}$), and carbon lost via dissolved organic transport (DOC). Many of these parameters are specific to global vegetation types: grasslands (GRA), evergreen forests (EF), shrublands (SHR), mixed forests (MF), and others (e.g., cropland, desert, wetland, and savanna).

before bootstrap resampling), the bootstrapped results (GPP$_{lit-group}$) are higher and closer to the GPP$_{Rs}$ (Supplementary Fig. 5). This suggests that older remote sensing approaches may underestimate sub-pixel spatial heterogeneity, and do not fully account for understory production[24] or belowground C allocation[25]. Second, products such as FLUXCOM are produced from eddy covariance measurements that are themselves spatially biased[26]. Furthermore, these measurements do not account for all carbon loss pathways or long-term $CO_2$ fertilization effects[9], and probably underestimate GPP in the highly-uncertain tropics[9,27], as well as in managed and fertilized croplands[28] where there are limited measurements to parameterize FLUXCOM. Finally, there are substantial uncertainties and mismatches in the algorithms that partition towers' net ecosystem exchange into GPP and respiration (Supplementary Table 3)[29], and also mismatches between these respiration estimates with direct measurements of $R_S$ (Table 2 and Supplementary Table 3).

Conversely, it is possible that Rs$_{lit}$ estimates are biased consistently high (Table 2). One important factor may be that $R_S$ data are less diverse than those of GPP, with almost all Rs$_{lit}$ ultimately deriving from a large but single global database of thousands of small-scale studies using generally similar methods[18]. This database is based on published data of annual fluxes, most of which are extrapolated (to an annual flux) from sporadic daytime measurements made at widely varying intervals, which might introduce bias[30]. Nevertheless, when additional newly published daily time scale in situ measurements were included to parameterize the RF models, global $R_S$ was predicted to be 93 Pg C yr$^{-1}$, very close to Rs$_{lit}$ (Fig. 1). Finally, the local- and/or large-scale models used to upscale measured $R_S$ temporally and spatially may not accurately represent soil moisture responses (e.g., due to hysteresis effects) because of its confounding effect with temperature[31].

A common potential problem affecting large-scale estimates of both GPP and $R_S$ concerns spatial coverage and representativeness of the terrestrial land surface and climate space[26]. GPP and

$R_S$ measurements have differing tradeoffs in this regard. The former is characterized by a spatially complete and large measurement domain (hundreds of m$^2$ to km$^2$, depending on the eddy covariance tower or pixel), but also nontrivial measurement uncertainties (e.g., the algorithms used to calculate GPP from the measured net flux). By contrast, $R_S$ is upscaled from spatially small (~1 m$^2$) but locally accurate chamber measurements dispersed in time that are, however, with better global coverage. Sites in both FLUXNET and SRDB are biased (Supplementary Fig. 6) towards the mid-latitudes of the northern hemisphere[32,33]. Both global GPP and $R_S$ are thought to be dominated by fluxes from highly-productive tropical forests, where eddy covariance towers are scarce and measurements, particularly uncertain[26,34]. Many of these factors could in theory produce systematic biases in the measurement and scaling of both GPP and $R_S$[9,35].

In addition, estimates of GPP$_{lit}$ and Rs$_{lit}$ have varied among studies (see Supplementary Fig. 8 and refs. [15,36]), reflecting methodological and technological differences, but uncertainty in these estimates have remained high (Supplementary Tables 1, 3); see also ref. [37]. We highlight that more recent GPP estimates have tended towards higher estimates but still with high uncertainty. There is also a temporal disparity when comparing literature estimates: while GPP$_{lit}$ and Rs$_{lit}$ cover a similar period overall (1980–2020), most GPP$_{lit}$ values are centered between 2000 and 2010, but a majority of Rs$_{lit}$ occurs between 1985 and 1995. If GPP$_{lit}$ and Rs$_{lit}$ are weighted equally by time (i.e., aggregated by the same breakpoints before bootstrap resampling, Supplementary Fig. 8), bootstrapped GPP$_{lit-agg}$ and Rs$_{lit-agg}$ are closer to GPP$_{Rs}$ and Rs$_{GPP}$ (by ~10 Pg C yr$^{-1}$, Supplementary Fig. 8), although significant disparities remain. Furthermore, when considering the temporal coverage and changing methods for GPP, we found that the gaps between carbon-cycle flux collected from the literature (GPP$_{lit}$ and Rs$_{lit}$) and the results implied by the other fluxes (GPP$_{Rs}$ and Rs$_{GPP}$) decreases, but still significantly differed from each other (P < 0.01, Supplementary Fig. 9).

**Perspective view.** How could we address these discrepancies and close the terrestrial C budget once and for all? The distribution of our GPP$_{Rs}$ and Rs$_{GPP}$ results is driven by a few key variables (Tables 1, 2), some of which are relatively rarely measured. These include the ratio of root respiration to total autotrophic respiration[38]; the ratio of root respiration to total soil respiration, and the ratio of autotrophic respiration to GPP; those data came from sites covering a similar range compared with global GPP, but lack measurements for regions with low photosynthesis (Supplementary Fig. 7). Acquiring (via field measurements or other approaches) additional constraints on these ratios may be a particularly fruitful way to resolve the inconsistencies identified in this study. For example, increasing numbers of studies have separated $R_S$ into its autotrophic and heterotrophic components in the last decade, enabling large-scale heterotrophic respiration synthesis efforts upscaling global estimates[16]. Recent studies have shown that $R_S$ are relatively less measured in low-productivity regions, arctic regions, and Tibetan Plateau, and that this uneven spatial distribution of data may create large uncertainties when scaling up and estimating global $R_S$[33,39], inferring GPP from Rs$_{lit}$ and inferring $R_S$ from GPP$_{lit}$ (Table 1) also show that $R_{root}$:$R_S$ and $R_{root}$:$R_A$ measurements from the desert, wetland, cropland, and savanna are key variables to close the gap between productivity and respiration fluxes in the global terrestrial carbon cycle. In addition, arctic regions and the Tibetan Plateau store a large amount of organic matter and are experiencing fast climate change. In the future, increasing field measurements of $R_{root}$:$R_S$,

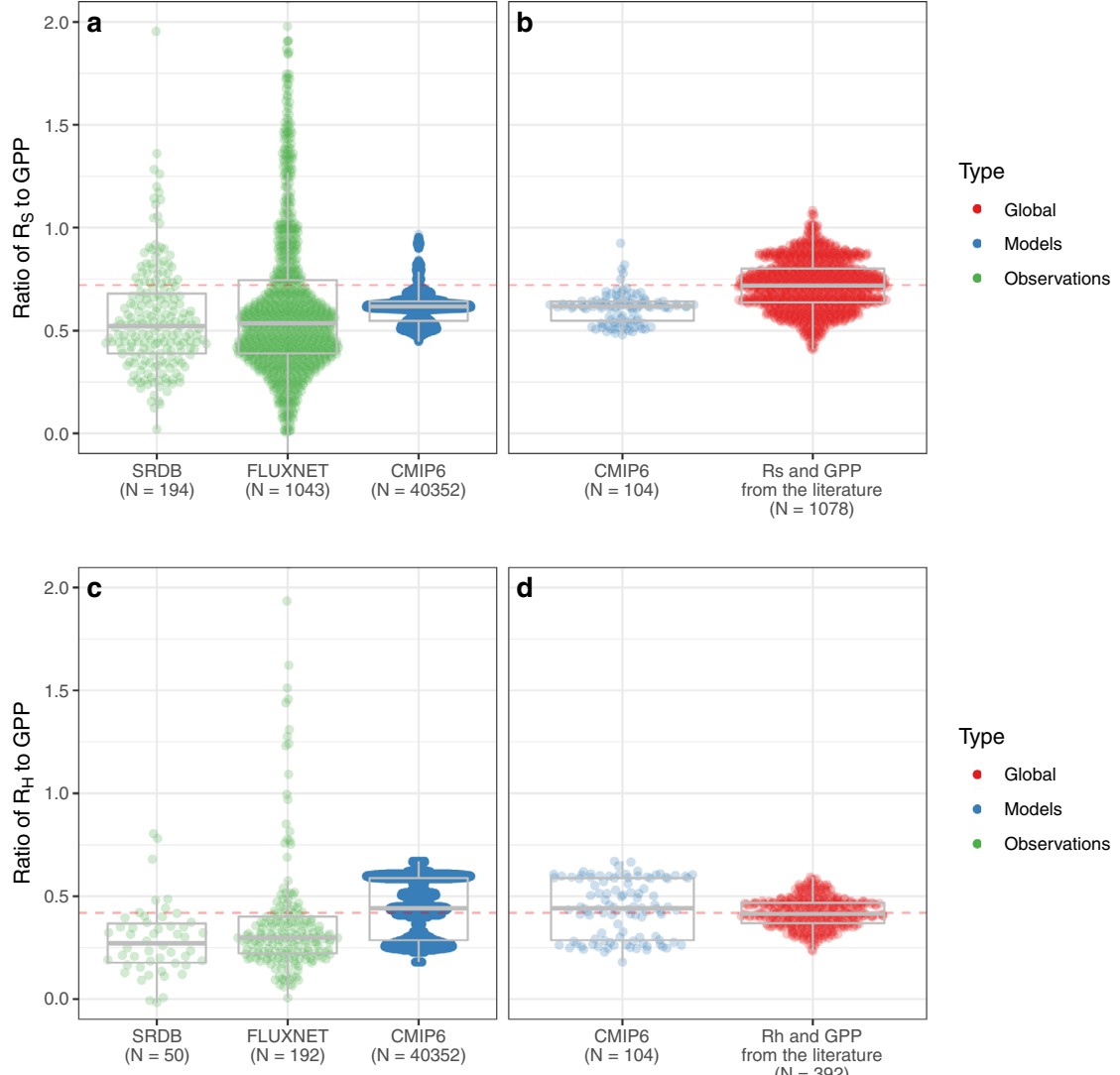

**Fig. 2 Observations, estimates, and model results of the ratio of soil respiration (R_S) or heterotrophic respiration (R_H) to gross primary productivity (GPP), at different spatial scales and from different sources. a** Observations, estimates, and model results of the ratio of $R_S$ to GPP at grid cell site-level; **b** Observations, estimates, and model results of the ratio of $R_S$ to GPP at a global scale; **c** Observations, estimates, and model results of the ratio of $R_H$ to GPP at grid cell site-level; **d** Observations, estimates, and model results of the ratio of $R_H$ to GPP at a global scale. Observational site-level data are from the global Soil Respiration Database (SRDB) and FLUXNET data (see Methods). The ratio of global $R_S$ and $R_H$ to global GPP is shown in red (and emphasized by the horizontal dashed lines), while results from the Coupled Model Intercomparison Project Phase 6 (CMIP6) at both the local grid cell site-level (values were extracted at coordinates corresponding to specific SRBD and FLUXNET sites) and global scale are shown in blue. Note that the odd distribution of the former results from the diversity of model ensemble realization used. Each point grouping is arranged distributionally, with overlaid box-and-whisker plots summarizing the mean, 25 and 75% quantiles, and extreme values. There are 16 models from CMIP6 with $R_H$ data; $R_S$ from CMIP6 models was calculated based on $R_H$ and $R_{root}:R_S$ ratio using a bootstrap approach.

$R_{root}:R_A$, and $R_A:$GPP, especially in low-productivity regions, arctic regions, and Tibetan Plateau is important to close the terrestrial carbon budget.

Here, we show large discrepancies between published estimates of global GPP and $R_S$, producing uncertainties that hamper our capacity to close the global C budget. Despite substantial efforts to understand carbon-climate feedbacks[2,35] in the last decades, changes to carbon uptake rates in response to climate change remain uncertain. Importantly, more recent GPP estimation methods—in particular, moving from MODIS-derived information to alternative measurements of plant photosynthetic activity (i.e., SIF)—seem to be closing the gap between our estimates of these two dominant terrestrial carbon fluxes. This is crucial, as, without accurate estimates of the largest terrestrial C fluxes, it will be impossible to correctly determine the land carbon sink and its

variability. Resolving the inconsistency between global GPP and $R_S$ is a necessary precondition for understanding the future of the global carbon cycle, and thus the possible future global climate change.

## Methods

**Carbon cycle terms and consistency**. This study explored the consistency of global gross primary productivity (GPP) and soil respiration ($R_S$) estimates in the global carbon (C) cycle. Terrestrial GPP is the photosynthetic gain of C by plants; soil respiration, the soil-to-atmosphere $CO_2$ flux, the sum of root respiration and heterotrophic respiration as measured at the soil surface, and represents carbon fixed by plants at some point in the past. While GPP and $R_S$ may diverge significantly at local scales and for short time periods, they should however be coupled to a degree consistent with our understanding of the C cycle[40]. Plant autotrophic respiration (including leaf and stem respiration, $R_{shoot}$, and root respiration, $R_{root}$) consumes part of GPP, and the remainder is termed net primary productivity (NPP). Parts of NPP are consumed by heterotrophs ($R_H$) and herbivores ($C_{herb}$),

---

**Table 2 Summary of uncertainties and possible biases: factors that might explain why gross primary production (GPP) would be biased low, and/or soil respiration ($R_S$) too high.**

| Possibilities for $R_S$ are biased too high | Possibilities for GPP are biased too low |
|---|---|
| 1) $R_S$ data are less diverse than those of GPP, with almost all $Rs_{lit}$ ultimately deriving from a large but single global database[18].<br>2) Tropical and subtropical forests are greatly under-sampled[52].<br>3) Jian et al.[39] showed that uneven distribution of $R_S$ sites may cause overestimation of global $R_S$ by ~6 Pg C yr$^{-1}$.<br>4) In situ Rs measurements may not be representative of Rs at ecosystem-scale[53, 54].<br>5) Rs cannot be measured directly at the ecosystem scale or using remote sensing, and we must upscale in situ measurements[14, 15, 23, 55].<br>6) Models do not have a clear mechanistic representation of Rs (as compared with GPP)[14, 15, 23, 55]. | 1) Satellite data algorithms and thus products have significant uncertainties (e.g., LAI and PAR conversion efficiency, $\varepsilon$)[7, 56-62].<br>2) Remote sensing may not fully account for understory production[24] or belowground C allocation[25].<br>3) GPP is probably underestimated in the tropics[9, 27], as well as in managed and fertilized croplands[28].<br>4) There are totally more than 900 flux tower sites worldwide (https://fluxnet.org/sites/site-summary/), but they are not evenly distributed, with some ecosystem types (e.g., tropic forests) less represented[63] (Supplementary Fig. 5).<br>5) Lack of $R_{root}$: $R_A$ ratio data for low photosynthesis productivity region (Supplementary Fig. 7d) |

---

burned in fires ($C_{fire}$), exported as dissolved organic carbon (DOC), or returned to the atmosphere by plants' biogenic volatile organic compound emissions (BVOC). The remainder comprises long-term carbon storage–the terrestrial carbon sink ($C_{sink}$). Theoretically, if we know how GPP is partitioned at each of these steps, we can produce an estimate of the $R_S$ implied by a GPP value (here termed $Rs_{GPP}$) at site or global scales; a similar process can be used to derive GPP from $R_S$.

**Data sources.** Global $R_S$ and GPP were collected from published literature. We collected 23 global $R_S$ estimates (Supplementary Table 1) from published articles, the majority of which upscaled site $R_S$ measurements based on a global database[41]. Approximately 100 scientific manuscripts estimated global GPP, and we used the following criteria to determine whether the GPP estimate should be included: (1) the GPP year (or middle year if GPP was averaged across a period, Supplementary Table 2) was after 1980; (2) GPP was estimated from satellite remote sensing products or upscaled from global flux data (as opposed to process-based modeling). With those criteria, 49 GPP estimates from published articles were used in this study (Supplementary Table 2).

Our primary source of global NPP estimates was a literature survey[42] that compiled 251 global NPP estimates. We noticed that there are several extreme NPP values within the dataset, we thus detected outliers using R, whatever an NPP estimate above 75% quantile + 1.5 interquartile range or below 25% quantile—1.5 interquartile range were considered as outliers. After outliers were removed, total 237 global NPP estimates were used in this study (Supplementary Fig. 1), similar to GPP. $C_{herb}$, $C_{fire}$, $C_{sink}$, DOC, and BVOC emissions were also collected from published literature (Supplementary Table 4). Ratios of root respiration to autotrophic respiration ($R_{root}$:$R_A$), autotrophic respiration to GPP ($R_A$:GPP), and root respiration to total soil surface respiration ($R_{root}$:$R_S$) were gathered from values in the global soil respiration database (SRDB[18]). Additional $R_{root}$:$R_A$ ratio data were collected from a literature search (Supplementary Table 5). We used the ISI Web of Science for all literature searches.

**Site-level data.** A number of site-specific data were used (the results of which appear in Fig. 1). The $R_S$:GPP ratio was computed based on observational data reported in the SRDB. To broaden the sources of available data for this analysis, we also used the FLUXNET-SRDB data combination from ref. [43]. Briefly, Tier 1 FLUXNET2015 data were downloaded 30 January 2017 from http://fluxnet.fluxdata.org/data/fluxnet2015-dataset/ and filtered for quality (NEE_VUT_REF_QC ≥ 0.5). FLUXNET GPP was linked to an SRDB $R_S$ measurement if both measurements occurred within 5 km, in the same vegetation type, and in the same year. We realized that if a land conversion occurred in the last decades, $R_S$ will not be in equilibrium with GPP making the Rs:GPP ratio incorrect, however, we believe this do not introduce an important bias because (1) usually $R_S$ and GPP are reported from the same study in SRDB, and thus land use and measurement year are the same; and (2) $R_S$:GPP ratio from SRDB are similar as that from FLUXNET (Fig. 2). This part of the analysis used eddy covariance data acquired and shared by the FLUXNET community, including these networks: AmeriFlux, AfriFlux, AsiaFlux, CarboAfrica, CarboEuropeIP, CarboItaly, CarboMont, ChinaFlux, Fluxnet-Canada, GreenGrass, ICOS, KoFlux, LBA, NECC, OzFlux-TERN, TCOS-Siberia, and USCCC.

**CMIP6 data processing.** Monthly historical GPP, heterotrophic respiration ($R_H$), and autotrophic respiration ($R_A$) outputs were obtained for the 16 models (104 model × ensemble combinations) currently available under the Coupled Model Intercomparison Project, version 6 (CMIP6[20]), from the Earth System Grid Federation archive (https://esgf.llnl.gov/, accessed February 23, 2020). But there are only two models have root respiration, therefore, we estimated root respiration of all CMIP6 models based on $R_A$ and Rroot:$R_A$ ratio (Supplementary Fig. 3). To calculate the annual $R_S$ and $R_H$ to GPP ratio, monthly outputs were processed using CDO 1.9.8[44] and R to obtain a global annual time series of C flux, weighted by land

area and the number of days in each month. This mean flux rate was converted to a total global flux by multiplying by the total land area and the number of seconds in a year, calculating $R_S$ as the sum of heterotrophic respiration and root respiration. To be consistent with the SRBD and FLUXNET observations, only data from those 1043 FLUXNET sites (Fig. 2) were extracted, the mean CMIP6 $R_H$ and $R_S$ to GPP ratio was calculated using flux data from 2005 to 2014.

For the ecosystem-scale CMIP6 analysis, we used monthly GPP, heterotrophic respiration, and root respiration outputs from 16 models. These were extracted at latitude and longitude coordinates corresponding to specific SRBD and FLUXNET sites. The total annual fluxes (weighted by days in a month) were used to calculate the average $R_S$ to GPP ratio from 2005 to 2014 at each coordinate. The final results consist of ratios at 362 latitude and longitude coordinates for 104 model × ensemble combinations. All CMIP6 processing code is available in the main repository at https://github.com/PNNL-TES/GlobalC.

**GPP implied by $R_S$ (GPP$_{Rs}$).** In the past decades, global $R_S$ rates have generally been estimated by upscaling site $R_S$ measurements (producing values here termed $Rs_{lit}$, meaning "$R_S$ estimates from literature"). We collected and summarized these estimates from published articles (Supplementary Table 1, $n = 23$); approximately half also reported $R_S$ 95% confidence interval or standard deviation ($N = 10$) and a rate of change during the study period ($N = 8$). The reported $R_S$ values ranged from 68 to 109 Pg C yr$^{-1}$, with an average of 85.4 Pg C yr$^{-1}$.

Some studies also separated $R_S$ into its heterotrophic ($R_H$) and root respiration ($R_{root}$) source fluxes; the resulting $R_{root}$:$R_S$ ratios have been compiled into the SRDB-V5[18] (Supplementary Fig. 2c). We used all of these $R_{root}$:$R_S$ ratios from SRDB-V5, in total 911 separate records between 0 and 1.0. These covered nine vegetation types, but the majority were from forest, grassland, cropland, and shrubland; all other vegetation types (desert, wetland, and savanna) had only 49 samples combined (Supplementary Fig. 2c).

Autotrophic respiration is made up of aboveground ($R_{shoot}$) and belowground ($R_{root}$) components. Many studies have separated $R_A$ into $R_{root}$ and $R_{shoot}$ (Supplementary Fig. 3 and Supplementary Table 5), and thus $R_{root}$:$R_A$ ratio and $R_{root}$:$R_{shoot}$ ratio can be calculated. GPP can be calculated (GPP$_{Rs}$, Supplementary Fig. 1 and Eqs. 1–3) from the $Rs_{lit}$ estimates according to $R_{root}$:$R_S$ ratio (RC), $R_{root}$:$R_{shoot}$ ratio (data from both the SRDB and an additional literature search, Supplementary Table 5) and NPP.

We then compared the GPP$_{Rs}$ with GPP from publications in past decades (i.e., GPP$_{lit}$) to determine the consistency between the GPP$_{lit}$ and GPP$_{Rs}$. The following equations were used to calculate GPP$_{Rs}$, i.e., the GPP implied by $Rs_{lit}$:

$$R_{root} = Rs_{lit} \times R_{root}:R_S \, ratio \tag{1}$$

$$R_{shoot} = R_{root} \times R_{shoot}:R_{root} ratio \tag{2}$$

$$GPP_{Rs} = NPP + R_{root} + R_{shoot} \tag{3}$$

**$R_S$ implied by GPP (Rs$_{GPP}$).** GPP has been estimated based on both remote sensing, FLUXNET data, and atmospheric inversions (Supplementary Table 2). We collected 49 such estimates from published articles; only 11 of these estimates reported the corresponding SD, and 14 reported corresponding temporal trends (Supplementary Table 2). The reported GPP estimates were from 1980 to 2015 and ranged from 100.2 to 167.0, with an average of 120.7 Pg C yr$^{-1}$.

GPP can be separated into NPP, $C_{herb}$, $C_{fire}$, $R_A$, DOC, BVOC, and $C_{sink}$. Our global NPP source was a previous meta-analysis[42], with outlier (outside 1.5 times the interquartile range above the upper quartile and below the lower quartile) removed, resulted in 237 estimates averaged 56.2 ± 9.6 Pg C yr$^{-1}$. After subtracting carbon consumed by herbivores, fire, and the land sink from NPP, global $R_H$ can be

estimated ($R_H = NPP - C_{herb} - C_{sink} - C_{fire} - DOC - BVOC$, Supplementary Fig. 1 and Supplementary Table 4).

The precise chain of reasoning and computation was as follows. The difference between GPP and NPP is $R_A$, meaning that an $R_A$:GPP ratio was required to estimate $R_A$ based on GPP (Eq. 4). The $R_A$:GPP ratios used in this study were from two sources: (1) a literature search that produced 123 $R_A$:GPP ratio estimates[45–48]; and (2) an additional 123 $R_A$:GPP ratio estimates from SRDB-V5. These $R_A$:GPP ratios covered nine vegetation types, mainly from forest and grassland; all the other vegetation types (cropland, wetland, and tundra) only had 14 samples combined (Supplementary Fig. 4). $R_A$ can also be calculated by subtracting NPP from GPP (Eq. 5), and calculated $R_A$ was very similar when computed by the above two methods. We used the average $R_A$ from these two methods.

In turn, $R_A$ consists of root respiration ($R_{root}$) and shoot respiration ($R_{shoot}$), and thus $R_{root}$:$R_A$ and $R_{shoot}$:$R_A$ ratios are required to calculate $R_{root}$ and $R_{shoot}$ from $R_A$. The $R_{root}$:$R_A$ ratios used in this study were from two sources: (1) 35 $R_{root}$:$R_A$ estimates from 28 literature studies (Supplementary Table 5); and (2) an additional 94 estimates from SRDB-V5. The $R_{root}$:$R_A$ values covered seven vegetation types (Supplementary Fig. 3), mainly from forests; all other vegetation types (cropland, savanna, grassland, and wetland) had only 18 samples.

Finally, starting with the GPP_lit values, and using NPP, $R_A$:GPP, $R_{root}$:$R_A$, and $R_{shoot}$:$R_A$, GPP can be separated into $R_H$, $R_{shoot}$, and $R_{root}$ and thus the implied global $R_S$ calculated (Rs_GPP; lower panel in Supplementary Fig. 1 and Eqs. 4–9 below). We then compared this Rs_GPP with Rs_lit to determine their consistency.

$$R_A = GPP_{lit} \times R_A{:}GPP \quad (4)$$

$$R_A = GPP - NPP \quad (5)$$

$$R_H = NPP - C_{sink} - C_{fire} - C_{herb} - DOC - BVOC \quad (6)$$

$$R_{root} = R_A \times R_{root}{:}R_A \quad (7)$$

$$R_{shoot} = R_A \times R_{shoot}{:}R_A \quad (8)$$

$$Rs_{GPP} = R_{root} + R_H \quad (9)$$

**Bootstrap resampling.** A critical factor is uncertainty that compounds at each step in this process. We used a bootstrap resampling approach to estimate GPP_Rs and Rs_GPP, as the sample size of each step is different, and many of the input data do not follow a normal distribution (Supplementary Figs. 1–5). For each bootstrap sample, we first generated a new estimate of GPP or $R_S$ by sampling from the published data (Supplementary Tables 1, 2, and 4, 5). We evaluated four different resampling methods, differing in how they treated the presence and absence of errors associated with each flux estimate. Method 1 did not use error information (i.e., any error estimate associated with each published $R_S$ or GPP value) when resampling. Methods 2–4 used errors but handled missing values differently. Method 2 replaced missing errors with values calculated from the median coefficient of variability (CV) of non-missing values; method 3 replaced missing errors with values calculated from the maximum CV across the dataset; and method 4 set missing errors to zero. We used method 3 in the main analysis, which is the most conservative (produces the widest distribution for both $R_S$ and GPP; cf. Supplementary Fig. 10).

In addition, a random value for each partitioning coefficient (e.g., above- to belowground autotrophic respiration ratio or herbivory fraction) was used in each bootstrap sample; note that errors are seldom reported for these data, and so were not considered here. We separated the $R_{root}$:$R_S$, $R_{root}$:$R_A$, and $R_A$:GPP ratios by vegetation type, weighted by global vegetation area (from the IGBP vegetation land classification, https://climatedataguide.ucar.edu/climate-data/ceres-igbp-land-classification). Starting from the randomly-drawn $R_S$ or GPP value, and randomly-drawn partitioning coefficients, the resulting $R_S$ or GPP was then calculated following Eqs. 1–9 described above.

**Variable importance analysis.** As noted above, many variables related to C partitioning were used to derive GPP from $R_S$ (Eqs. 1–3) or to derive $R_S$ from GPP (Eqs. 4–9). To determine the relative contribution of each variable to the overall distributional uncertainty, as well as the sensitivity of the estimate to that variable, we fixed each variable (e.g., NPP) in turn to the median of all its observations. All other variables were randomly drawn, as normal, in the bootstrap process, and the output variable (GPP_Rs or Rs_GPP) mean and distribution were calculated. We then compared the output variance with the result when no variables were fixed, i.e., that shown in Fig. 1, to determine the importance of each variable: larger decreases in output variance when a particular parameter was fixed to be constant, imply greater importance for this parameter.

**Representativeness analysis.** We connect the $R_{root}$:$R_S$, $R_{root}$:$R_A$, and $R_A$:GPP sites with external global GPP data from FLUXCOM (https://www.fluxcom.org/, last accessed on 2021/06/22) through latitude and longitude to obtain mean GPP between 2001 and 2015. We then compared the GPP of sites used in this study with the global GPP (spatial resolution of 0.5°) to test the representation of the sites (Supplementary Fig. 7).

**Overlap calculation.** We calculated the overlap between the GPP_lit distribution and the distribution of GPP_Rs to quantify the agreement between GPP_lit and GPP_Rs. If a sample was not significantly different from a normal distribution (based on a Shapiro–Wilk test in R), we used a normal distribution with sample mean and variance to approximate the distribution; if a sample was significantly different from a normal distribution, we used a numerical approximation based on linear interpolation (*approxfun* in R) to approximate the distribution's probability density function. We then calculated the intersection point of these probability density functions, as well as the proportion of each curve that overlapped with the other using a trapezoidal rule numerical integration. Finally, we sampled each approximated distribution for the original number of GPP or $R_S$ values. With these samples, a two-sample Welch's *t*-test (*t.test* with *var.equal = FALSE* in R) was performed to determine if the means of the two distributions differed significantly.

**Global soil respiration modeling.** Following a similar approach as Jian et al. (2018)[23], measurements from a global daily soil respiration database (DGRsD) and nine environmental factors (i.e., nitrogen deposition, monthly precipitation, monthly air temperature, soil bulk density, soil organic carbon, soil clay percentage, aboveground biomass, belowground biomass, and Enhanced Vegetation Index, details please see supplementary Table 6) were used to build Random Forest (RF) models for each month. Only $R_S$ measurements with no field manipulation were used, totally 27,214 samples were separated into two datasets, 80% of samples were used to train the models, and the rest 20% were used to test the model performance. The results showed that the RF models can explain ~66% $R_S$ variability, and the performance is consistent with both training and validation datasets. $R_S$ for each month with a spatial resolution of 0.1° were predicted by the RF models, estimated monthly $R_S$ were then summarized to estimate global annual $R_S$. Permanent ice sheets and bare soils were removed according to the MODIS landcover map[49].

**Other statistical analyses.** All analyses were conducted using R 3.6.1[50]. Bootstrap means were compared using a two-sided Student's *t*-test. A one-sided, nonparametric Wilcoxon rank-sum test with continuity correction was used to compare $R_S$ to GPP ratios calculated from global estimates, the SRDB, and CMIP6 outputs.

## Data availability
The data to support all the analysis in this study have been deposited in the GitHub repository [https://github.com/PNNL-TES/GlobalC/] and zenodo [https://doi.org/10.5281/zenodo.5900964][51].

## Code availability
The code to reproduce all the results in this study have been deposited in the GitHub repository [https://github.com/PNNL-TES/GlobalC/] and zenodo [https://doi.org/10.5281/zenodo.5900964][51].

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

## Acknowledgements

This research was supported by the second Tibetan Plateau Scientific Expedition and Research Program (STEP) (No. 2019QZKK0603), the Strategic Priority Research 482 Program of the Chinese Academy of Sciences (No. XDA20040202), and the Pacific Northwest National Laboratory is operated for the US Department of Energy, Office of Science, Biological and Environmental Research as part of the Terrestrial Ecosystem Sciences Program by Battelle Memorial Institute under contract DE-AC05-76RL01830. R.V. was supported by the NASA Carbon Monitoring System 80NSSC18K0179. A.G.K. was supported by NASA NNH16ZDA001N-IDS and by NSF DEB-1942133. This work used eddy covariance data acquired and shared by the FLUXNET community (see Methods).

## Author contributions

J.J. conceived this study, and with A.N.S. and B.B.-L. designed the primary analysis. K.D. processed and analyzed CMIP6 data. A.S. conceptualized and coded a number of the

numerical calculations. D.H. processed the Enhanced Vegetation Index data, was involved in the random forest modeling to predict global monthly soil respiration, and generated NetCDF data for the global soil respiration as well as generated the global soil respiration map. V.B., A.G.K., M.S., M.T., D.H., and R.V. provided feedback and insights in all phases. J.J. and B.B.-L. wrote the manuscript in close collaboration with all authors.

## Competing interests

The authors declare no competing interests.
