## [Peer Review File · Nature Communications]

Historically inconsistent productivity and respiration fluxes in the global terrestrial carbon cycleReviewers' Comments:

Reviewer #1:

Remarks to the Author:

Jian et al follow two complementary lines of evidence to show that estimates of global annual GPP and soil respiration (R_s) in the current scientific literature are inconsistent with each other. The authors did an extensive literature research on annual estimates of global GPP and R_s derived from different approaches using satellite observations (and excluding those that are based on process-based modelling where the carbon fluxes are constrained by each other intrinsically). Similar literature estimates of NPP, of partitioning ratios involving autotrophic (total, roots, shoots) and heterotrophic respiration in different vegetation types as well as of other important carbon pathways (fire, VOCs, DOC, herbivores, land sink) are combined in a bootstrapping approach to derive estimates of GPP based on literature-reported R_s and viceversa. The bootstrapped distributions overlap only marginally with the literature-derived distributions and they are highly unlikely to derive from the same population. In a complementary approach site-level chamber and eddy-covariance measurements are found to be inconsistent with literature-reported R_s /GPP ratios. The same is true for R_s /GPP ratios from CMIP6 ensemble members (at pixel-level and globally). The authors argue that most likely current GPP estimates are biased low.

The main finding of globally inconsistent GPP and R_s is indeed noteworthy and definitely of interest to the scientific community. Most studies focus on one terrestrial carbon flux only and compare estimates from a variety of approaches, while the main novelty of this study is a systematic inter-comparison across approaches and fluxes that tries to close the budget. This is indeed an absolutely non-trivial task given the many sources of biases and uncertainties in the individual data sources employed. Each of the two approaches individually needs several important assumptions and/or is based on data with very limited representativeness. Therefore, the presentation of two complementary lines of evidence strongly supports the main result and increases its credibility.

My expertise does not cover R_s , chamber measurements and partitioning ratios, I will therefore mostly refrain from commenting on these aspects (or comment/ask questions from the perspective of someone not very familiar with such aspects as in the 2nd major point below).

As a major point of improvement I see the evaluation and discussion to what extent the literature estimates of GPP and R_s (but eg also C_{fire}) are comparable as they cover almost systematically different temporal periods. Also the extent to which the literature-based partitioning ratios are applicable and representative globally needs clarification in my opinion (see further explanation below).

The authors delivered a very balanced discussion on whether GPP is rather biased low or whether R_s is biased high which is appreciated. However, given the variety of GPP approaches collected from the literature compared to the R_s estimates which are largely all based on the same collection of in situ observations, in my eyes the hypothesis that mainly a low bias in GPP causes the observed inconsistency between GPP and R_s is not sufficiently supported with the arguments in ll. 173-180. Otherwise the discussion and presentation are very balanced and comprehensive. The manuscript is clearly written and structured, and the methods are adequate and also described in sufficient detail to be reproducible. I suggest major revision.

1) To me, the weakest point in this manuscript is the discussion of the influence of temporal representativeness, scales and resolution:

Figure S7 and Tables S1, S2 and S3 show that $R_{s, lit}$ rather covers the period before the mid 1990s, while GPP_{lit} rather refers to later years mostly between 2000-2010 and that there is little temporal overlap. Given the positive trends in both R_s and GPP, where $trend(GPP) > trend(R_s)$ (tables S1 and S2 as well as other published literature eg on trends in modelled Trendy fluxes), I wonder what the effect of this almost systematic difference in temporal representativeness will be on the overall inconsistency between GPP and R_s . Can one assume that GPP is already biased high compared to R_s as a result of representing later years, and that the effect of inconsistency between GPP and R_s might be even

stronger? Does the magnitude of the trends matter at all compared to the magnitude of the inconsistency? What are the trends of literature GPP and Rs given in tables S1 and S2 used for? Also, in the estimated carbon emissions by fire in table S3 later years are rather under-represented compared to earlier decades and should be discussed.

The authors slightly touch upon the effect of temporal resolution in the case of the random forest modelling of Rs. Otherwise I read that the measurement time and frequency do not cause any significant effect on annual Rs (Jian et al. 2020) at site-level. This is surprising to me given 'sporadic daytime measurements made at widely varying intervals' (l.167), and I wonder how this compares to the associated Fluxnet observations which are supposed to be rather continuous during a day and a year?

2) I am not familiar with the datasets of partitioning ratios, but these measurements are notoriously scarce and obtain a very high variable importance in the bootstrapping. I therefore would have expected a more comprehensive discussion of their representativeness and potential influence on the results. For example, to what extent are climatic gradients within vegetation types represented in this database and should they be? Also, how representative are they on annual time scales (compared to the point in a season when they were measured)? Are those measurements obtained in field or lab conditions? Further, the grouping of vegetation types seems inconsistent between partitioning ratios, particularly crops and grasslands are treated individually in one ratio and combined into the 'other'-group in another ratio. In Root:Rs, there is a vegetation type called 'agricultural' while there is a group 'crop' in the other ratios. What is the difference? In Root:Rs, shrubs are treated both individually and combined into 'other'?

3) In the paragraph in lines 114-130, both the site-level Fluxnet and SRDB-derived Rs:GPP ratios as well as those obtained from CMIP6 models are found to be inconsistent with literature derived Rs:GPP ratios. The estimates from CMIP6 and Fluxnet/SRDB differ also quite strongly magnitude-wise, how consistent are those statistically with each other?

Minor:

Table 1 caption l.232: 'NPP was responsible for 43.9% of the total variance when inferring Rs from GPP.' This number is inconsistent with the number given in the table.

Reviewer #2:

Remarks to the Author:

The objective of the paper is to synthesize, from different sources (in situ data, remote sensing modeling etc..) our knowledge on global GPP and soil respiration Rs. Making some assumptions on link between both terms, it is then possible to evaluate how these independent estimations are or not in agreement. This is an interesting topic as few attempts have been done to compare these estimations. The paper makes a comprehensive compilation of all existing data and describes all the sources of uncertainties that affect both Ts and GPP that can explain the discrepancy found between the two terms. However, the main conclusion of the paper is that the large discrepancy between Rs and GPP is more likely related to an underestimation of GPP instead of an overestimation of respiration. I find this conclusion very speculative and I am not really convinced by the arguments given to support it. The first argument is the fact that Ts would not be biased because the sources of uncertainties listed in table 2 can induce under or over estimation of small amplitude and then should compensate. However, as nothing is given quantitatively on the amplitude of these different biases (in particular there is no argument to say that these biases are of small amplitude) there is no reason why positive and negative bias should be equivalent and then compensate!

The second argument is to say that the new estimate of Rs proposed in the study based on a RF algorithm and an updated global monthly database (supposed to limit the problem of temporal representativity of measurements in the SRDB database) find a global estimate of Rs of 93 PgC yr⁻¹,

higher than estimation from the literature and should provide a more robust estimate. However even if it is based on an updated database, it is mainly based on the same kind of data the same limitation than others databases especially on heterogeneity on the spatial coverage of such kind of data. So such study cannot be really considered as an independent estimate from previous ones and, should have the same bias, if exists, that others studies.. Another argument presented in the case of R_s :GPP ratio explicitly given in the SRDB database or estimated from combination of SRDB with FLUXNET is to explain that there is probably no bias related to spatial extend of data since weighting (or not) the measurements by vegetation areas globally gives similar results. But it just proves that values from underrepresented vegetation type is not very different for those that are overestimated. But if sites on vegetation from which there is only few measurements are not well representative of the whole ecosystem we can still have large biases . We can also notice that if GPP is estimated from the in-situ R_s :GPP ratio of 0.54 and the new global R_s estimate of 93 GtC we find a global GPP value of 172 GtC which is even larger than estimation given from GPP_ R_{slit} !

In addition to sources of uncertainties from R_s that are mentioned in the manuscript there is the large spatial heterogeneity of respiration even at a very small spatial scale making chambers measurements not necessarily representative of a stand (in opposite to flux measurement that is more integrative). More important, R_s largely depend on the land use history of the site. This means that R_s :GPP ratio can be largely variable within a given ecosystem and climate condition not because of uncertainty in the measurement but just because of disequilibrium between GPP and heterotrophic respiration. Hence for instance an R_s :GPP ratio on cropland with soil largely depleted in carbon or on a reforested area can be probably around 0.2 whereas, on the opposite, R_s :GPP, in the same conditions, on a recently deforested cropland or on the beginning of a forest rotation can probably reach 0.7. So for me uncertainties in estimation of R_s is probably as large as estimation on GPP (even probably higher) and then it is very difficult to conclude from the information given in the paper anything about which one of the terms is the major cause of the discrepancies which limit the main message of the paper. Probably the best argument is that some recent methods for GPP estimations seems to be more in agreement with estimated GPP deduced from R_{slit} (which is in some way contradictory with the title which state that productivity and respiration fluxes are inconsistent), I discuss this point more in detail hereafter. But here again these new methods are not necessarily less uncertain than previous one predicting lower global GPP (for instance SIF data is very promising as it is a more direct proxy of GPP than NDVI but still very preliminary and then also very uncertain) .

One important point is that, in opposite to R_s estimation which is only based on direct measurement or estimation for flux tower, GPP can be estimated by different independent methods . This is unfortunately only briefly discussed in the paper and only shown in figure S5. This is however an important point. In particular looking to table S2 we can see that a large majority of estimates used in the study come from MODIS which gives all relatively similar estimations in the lower range of the estimated GPP. Then the mean GPP $_{lit}$ of 113 PgC given as reference calculated by averaging all the GPP estimates from literature is mainly dominated by MODIS estimations and then relatively low. For instance, if instead of averaging all the values we first average the values coming for each method (I.e NDVI, satellite driven models, FLUXCOM, FLUXNET, SIF, O18, IPCC, global carbon) and then averaging the mean value of each method to give a similar weight to each method independently of the number of available estimation the mean value is 130 GtC which is still lower than estimated from R_s but closer however. More generally, as briefly mentioned, some methods like O18 of SIF for instance gives estimates of 162 GtC and 147 GtC which are in agreement with GPP estimated from R_{slit} of 149GtC ! But as I mentioned previously, it doesn't prove that these values are more realistic than estimation based on remote sensing NDVI which are around 112GtC but at least it shows that the estimations of GPP and R_s are not necessarily fully inconsistent as claimed in the title. Likewise the fact to say that GPP is probably underestimation is not necessarily true for all the method considered So it would be important to have a deeper discussion on the different methods of estimation of GPP and not just look to the average of these estimations.

Concerning the discussion of R_s :GPP simulated by ESMs there is an important limit which is related to

the fact that only two versions of the same ESM model (CESM2) is considered in the study. Considering the large spread of modeled respirations in the different ESMs, it is very difficult to give a general conclusion on simulated R_s :GPP ratios in ESMs based only on a single model. Obviously I understand that it is related to the fact that only CESM2 report the partition of respiration within the different biomass compartments in the CMIP6 database. However there is some ways to see how CESM2 behave (in term of ratio between respiration and GPP) compared to others models. For instance, one can see how ratio of total R_a to GPP and R_h to GPP in CESM2 compare to others models. Probably, most of the models provide also the partition of biomass between the different compartments. Even if respiration coefficient can be different for the different compartments, weighting R_a with the relative biomass of each compartment could be a first guess of root respiration as well. So to be able to discuss what is the ratio of R_s to GPP from the models point of view it is very important that CESM2 could be compared to others ESMs to see if it is, for the different ratio of respiration to GPP, in the middle of models range or at one of the end members. In particular from what I understand in the text R_{a_root} :GPP is 21% and R_s :GPP 27-29% that would mean that R_h :GPP is only 6-8 % which seems unrealistic.

About site-level data it is mentioned that FLUXNET GPP was linked to an SRBD R_s if both measurements occurred within 5km in the same vegetation type and the same year. However, as I mentioned previously, there is another criteria that should be also taken into account which is the land use history of the site. Indeed if a land conversion occurred in the last decades, R_s will not be in equilibrium with GPP making the R_s :GPP ratio incorrect. This is obviously a difficult question as the site history is not necessarily reported. The fact that the estimated R_s :GPP ratio estimated is similar to value reported in the SRBD data (for which I guess both R_s and GPP are reported from the same site) seems to indicate that this not induce an important bias but this point should be considered carefully and discussed however.

In summary the paper make an interesting review of our existing knowledge on both global GPP and soil respiration. It shows in particular that there is still very large uncertainties in the different terms making R_s and GPP difficult to reconcile at global scale. However there is some assumptions or conclusions that are not very convincing especially the fact that the most likely "suspect" would be a too low estimate of GPP. Soil respiration measurement it, at least, as uncertain as GPP. Moreover, since there is several methods to estimate GPP, we can have an estimation of the possible range which is not the case for respiration for which only extrapolation from in-situ measurement are available and then few means to access its effective uncertainty. Some methodological aspect should also be revised. This is the case of evaluation of R_s :GPP ratio from ESM that should not only based on a single model. It is also important to discuss ratio of R_s :GPP in regard of the different methods for estimation of GPP.

Reviewer #3:

Remarks to the Author:

Summary:

The authors present a compilation of literature values for GPP and soil respiration (R_s) and show that they are not consistent with one-another. They find a mismatch; that either GPP is too small or R_s is too large. They conclude that the most likely scenario is that GPP is currently underestimated when considering potential biases of both global GPP and R_s estimates. They also show that ratios of R_s /GPP are highly variable across methods: 0.54 for site-level observations, 0.71 for global estimates, and 0.28 for modeled estimates.

General strengths:

The approach is novel, and to my knowledge, has not been done before. The compilation effort is nice. The GPP and R_s measurement communities are operating pretty independent of one another so this

comparison is extremely useful. The range of inconsistencies that the authors point out is troubling for the field.

General comments:

I previously reviewed this manuscript when the authors submitted to a different journal. I've compared my previous comments to this revised version and most of my questions have been answered in the new submission, therefore this review is unusually brief. They have added a discussion of why the Rs and GPP estimates may not reconcile which adds value to the paper in my opinion. I think this would make a nice contribution to Nature Communications in its current form.

Specific comments:

Line 137: ...implied by atmospheric 18O:16O ratios of CO₂...

The figure captions are a little hard to understand and could use a grammar check.

Figure 2: global CMIP6 results do not appear to be dark blue like the caption indicates. Perhaps the symbols are simply behind the grey box-and-whisker plot though.

Table 2: I don't find this table especially interesting to warrant including in the main text. It would be fine in supplemental.

Fig S5: Are atmospheric constraint estimates missing from this figure? What is FLUXNET isotope data?

REVIEWER COMMENTS

Reviewer #1 (Remarks to the Author):

Reviewer: Jian et al follow two complementary lines of evidence to show that estimates of global annual GPP and soil respiration (Rs) in the current scientific literature are inconsistent with each other. The authors did an extensive literature research on annual estimates of global GPP and Rs derived from different approaches using satellite observations (and excluding those that are based on process-based modelling where the carbon fluxes are constrained by each other intrinsically). Similar literature estimates of NPP, of partitioning ratios involving autotrophic (total, roots, shoots) and heterotrophic respiration in different vegetation types as well as of other important carbon pathways (fire, VOCs, DOC, herbivores, land sink) are combined in a bootstrapping approach to derive estimates of GPP based on literature-reported Rs and vice versa. The bootstrapped distributions overlap only marginally with the literature-derived distributions and they are highly unlikely to derive from the same population. In a complementary approach site-level chamber and eddy-covariance measurements are found to be inconsistent with literature-reported Rs/GPP ratios. The same is true for Rs/GPP ratios from CMIP6 ensemble members (at pixel-level and globally). The authors argue that most likely current GPP estimates are biased low.

Response: Thanks for the insightful summary of our manuscript. In the revision, we decide not to emphasize that low GPP is the reason for the inconsistent productivity and respiration fluxes in the global terrestrial carbon cycle. Instead, we note that either GPP could be biased low, or Rs estimates could be biased high (summarized in Table 2 in the revision).

Reviewer: The main finding of globally inconsistent GPP and Rs is indeed noteworthy and definitely of interest to the scientific community. Most studies focus on one terrestrial carbon flux only and compare estimates from a variety of approaches, while the main novelty of this study is a systematic inter-comparison across approaches and fluxes that tries to close the budget. This is indeed an absolutely non-trivial task given the many sources of biases and uncertainties in the individual data sources employed. Each of the two approaches individually needs several important assumptions and/or is based on data with very limited representativeness. Therefore, the presentation of two complementary lines of evidence strongly supports the main result and increases its credibility.

Response: We agree with this opinion about the importance of having two independent approaches to close the budget. We agree that this is an important novel contribution from our study.

Reviewer: My expertise does not cover Rs, chamber measurements and partitioning ratios, I will therefore mostly refrain from commenting on these aspects (or comment/ask questions from the perspective of someone not very familiar with such aspects as in the 2nd major point below).

As a major point of improvement I see the evaluation and discussion to what extent the literature estimates of GPP and Rs (but eg also C_{fire}) are comparable as they cover almost systematically different temporal periods. Also the extent to which the literature-based partitioning ratios are

applicable and representative globally needs clarification in my opinion (see further explanation below).

The authors delivered a very balanced discussion on whether GPP is rather biased low or whether R_s is biased high which is appreciated. However, given the variety of GPP approaches collected from the literature compared to the R_s estimates which are largely all based on the same collection of *in-situ* observations, in my eyes the hypothesis that mainly a low bias in GPP causes the observed inconsistency between GPP and R_s is not sufficiently supported with the arguments in ll. 173-180.

Otherwise the discussion and presentation are very balanced and comprehensive. The manuscript is clearly written and structured, and the methods are adequate and also described in sufficient detail to be reproducible. I suggest major revision.

Response: We agree with this opinion, which is shared by other reviewers also concerned about our argument about too low GPP. In the revision, we no longer emphasize that low GPP may be the reason for inconsistency between GPP and R_s ; instead, we now suggest that either GPP is biased low (Lines 167-181) or R_s may be biased high (Lines 183-194), and future efforts should be devoted to close the gap. We think this is a more appropriate conclusion that serves as a better guide for future research.

Reviewer: 1) To me, the weakest point in this manuscript is the discussion of the influence of temporal representativeness, scales and resolution:

Figure S7 and Tables S1, S2 and S3 show that $R_{s\text{lit}}$ rather covers the period before the mid 1990s, while GPP_{lit} rather refers to later years mostly between 2000-2010 and that there is little temporal overlap. Given the positive trends in both R_s and GPP, where $\text{trend}(GPP) > \text{trend}(R_s)$ (tables S1 and S2 as well as other published literature eg on trends in modelled Trendy fluxes), I wonder what the effect of this almost systematic difference in temporal representativeness will be on the overall inconsistency between GPP and R_s . Can one assume that GPP is already biased high compared to R_s as a result of representing later years, and that the effect of inconsistency between GPP and R_s might be even stronger? Does the magnitude of the trends matter at all compared to the magnitude of the inconsistency? What are the trends of literature GPP and R_s given in tables S1 and S2 used for? Also, in the estimated carbon emissions by fire in table S3 later years are rather under-represented compared to earlier decades and should be discussed.

The authors slightly touch upon the effect of temporal resolution in the case of the random forest modelling of R_s . Otherwise I read that the measurement time and frequency do not cause any significant effect on annual R_s (Jian et al. 2020) at site-level. This is surprising to me given ‘sporadic daytime measurements made at widely varying intervals’ (l.167), and I wonder how this compares to the associated Fluxnet observations which are supposed to be rather continuous during a day and a year?

Response: This is a great point, and we agree with this point about temporal representativeness; indeed, $R_{s\text{lit}}$ covers the period before the mid-1990s, while GPP_{lit} covers the period mostly between 2000 and 2010. We now analyzed how this temporal representativeness could affect the results. We first aggregate $R_{s\text{lit}}$ and GPP_{lit} by year, then use bootstrapping to get the distribution

of R_s . The results showed that, based on the aggregated R_s data, bootstrapped mean $R_{s\text{lit-agg}}$ is lower than that implied by the raw data ($R_{s\text{lit}}$), which is closer to the $R_{s\text{GPP}}$ (see figure below and Figure S8e in the revision). Meanwhile, based on the aggregated GPP data, bootstrapped mean $\text{GPP}_{\text{lit-agg}}$ is higher than that implied by raw data (GPP_{lit}), which is closer to the GPP_{R_s} (Figure S8f). Those additional analysis thus provide evidences that temporal representativeness as well as GPP estimation methods do affect the conclusion, but these distributions still significantly differed from each other ($p < 0.01$, see Figure S9 in the revision). We now discuss this in the revision (lines 196 – 215).

Reviewer: 2) I am not familiar with the datasets of partitioning ratios, but these measurements are notoriously scarce and obtain a very high variable importance in the bootstrapping. I therefore would have expected a more comprehensive discussion of their representativeness and potential influence on the results. For example, to what extent are climatic gradients within vegetation types represented in this database and should they be?

Response: That is correct: partitioning ratios are scarce, and while we tried hard to obtain these data, the available coverage is not ideal. In the revision, we present the comparison of mean GPP of $R_{\text{root}}:R_S$, $R_{\text{root}}:R_A$, and $R_A:\text{GPP}$ ratio sites vs. GPP around the globe (from FLUXCOM, averaged between 2001-2015, and scaled to spatial resolution of 0.5°). This new analysis showed that a lack of $R_{\text{root}}:R_A$ measurements from low photosynthesis production regions could result in low R_{SGPP} . We also discuss the potential bias related with this issue, please see figure below and Figure S7 and lines 219-232 in the revision.

Reviewer: Also, how representative are they on annual time scales (compared to the point in a season when they were measured)?

Response: Most $R_{\text{root}}:R_S$, $R_{\text{root}}:R_A$, and $R_A:\text{GPP}$ ratio data used in the study are from SRDB, which is an annual time scale database. All $R_{\text{root}}:R_S$ ratio data used in this study are from SRDB and in annual time scale. We also did a literature search to obtain more $R_{\text{root}}:R_A$, and $R_A:\text{GPP}$ ratio data, we went through all those articles to identify the temporal coverage. For the $R_{\text{root}}:R_A$

collected from studies, we added a column to specify the temporal coverage of $R_{\text{root}}:R_A$ data in Table S5 in the revision now. For $R_A:\text{GPP}$ data, we collected data from 4 articles (see reference below), and the temporal coverage of all values we used are annual time scale. To summarize, the overwhelming majority of all these data were measured over at least a full year.

Amthor, Jeffrey S., and Dennis D. Baldocchi. "Terrestrial higher plant respiration and net primary production." *Terrestrial global productivity* (2001): 33-59.

Piao, Shilong, Sebastiaan Luyssaert, Philippe Ciais, Ivan A. Janssens, Anping Chen, Chao Cao, Jingyun Fang, Pierre Friedlingstein, Yiqi Luo, and Shaopeng Wang. "Forest annual carbon cost: A global - scale analysis of autotrophic respiration." *Ecology* 91, no. 3 (2010): 652-661.

Ma, Siyan, Dennis D. Baldocchi, Liukang Xu, and Ted Hehn. "Inter-annual variability in carbon dioxide exchange of an oak/grass savanna and open grassland in California." *Agricultural and Forest Meteorology* 147, no. 3-4 (2007): 157-171.

Kinerson, R. S., C. W. Ralston, and C. G. Wells. "Carbon cycling in a loblolly pine plantation." *Oecologia* 29, no. 1 (1977): 1-10.

Reviewer: Are those measurements obtained in field or lab conditions?

Response: All data used were obtained in the field, we clarified this in the caption of Table S5.

Reviewer: Further, the grouping of vegetation types seems inconsistent between partitioning ratios, particularly crops and grasslands are treated individually in one ratio and combined into the 'other'-group in another ratio. In $R_{\text{root}}:R_s$, there is a vegetation type called 'agricultural' while there is a group 'crop' in the other ratios. What is the difference? In $R_{\text{root}}:R_s$, shrubs are treated both individually and combined into 'other'?

Response: It's true that the grouping of vegetation types is different for $R_{\text{root}}:R_s$ ratio (Figure S2), $R_{\text{root}}:R_A$ ratio (Figure S3), and $R_A:\text{GPP}$ ratio (Figure S4); we have to treat these somewhat differently because we have more $R_{\text{root}}:R_s$ ratio data but much less $R_{\text{root}}:R_A$ ratio and $R_A:\text{GPP}$ ratio data. For example, we do not have a separate grassland category for $R_{\text{root}}:R_A$ ratio because we would be only 2 data points in it, and so they were grouped into the "Other" group. Agriculture is the same thing as cropland, and we have resolved this name inconsistency (please see Figure S2). Shrubs should not be included in "Other"; this has been resolved in the figure caption (please see the caption of Figure S2 in the revision).

Reviewer: 3) In the paragraph in lines 114-130, both the site-level Fluxnet and SRDB-derived $R_s:\text{GPP}$ ratios as well as those obtained from CMIP6 models are found to be inconsistent with literature derived $R_s:\text{GPP}$ ratios. The estimates from CMIP6 and Fluxnet/SRDB differ also quite strongly magnitude-wise, how consistent are those statistically with each other?

Response: We used Wilcoxon method to test the differences between groups, and found that literature derived $R_s:\text{GPP}$ ratio is significantly higher than others, this was mentioned in the revision (Line 121-122).

Reviewer: Minor:

Table 1 caption l.232: 'NPP was responsible for 43.9% of the total variance when inferring Rs from GPP.' This number is inconsistent with the number given in the table.

Response: Thanks for pointing this out, resolved and we also carefully checked all numbers in the whole manuscript to avoid similar mistakes.

Reviewer #2 (Remarks to the Author):

Reviewer: The objective of the paper is to synthesize, from different sources (in situ data, remote sensing modeling etc..) our knowledge on global GPP and soil respiration R_s . Making some assumptions on link between both terms, it is then possible to evaluate how these independent estimations are or not in agreement. This is an interesting topic as few attempts have been done to compare these informations. The paper makes a comprehensive compilation of all existing data and describes all the sources of uncertainties that affect both T_s and GPP that can explain the discrepancy found between the two terms. However, the main conclusion of the paper is that a large discrepancy between R_s and GPP is more likely related to an underestimation of GPP instead of an overestimation of respiration. I find this conclusion very speculative and I am not really convinced by the arguments given to support it. The first argument is the fact that T_s would not be biased because the sources of uncertainties listed in table 2 can induce under or over estimation of small amplitude and then should compensate. However, as nothing is given quantitatively on the amplitude of these different biases (in particular there is no argument to say that these biases are of small amplitude) there is no reason why positive and negative bias should be equivalent and then compensate!

Response: Thanks for the overall positive opinion on this manuscript. We agree that we do not have enough evidence to support the argument that underestimation of GPP causes the inconsistency. In the revision, we do not emphasize low GPP, but rather highlight that either GPP is biased low (Lines 167-181) or R_s is biased high are possible (Lines 183-194).

Reviewer: The second argument is to say that the new estimate of R_s proposed in the study based on a RF algorithm and an updated global monthly database (supposed to limit the problem of temporal representativity of measurements in the SRDB database) find a global estimate of R_s of 93 PgC yr⁻¹, higher than estimation from the literature and should provide a more robust estimate. However, even if it is based on an updated database, it is mainly based on the same kind of data with the same limitation than other databases especially on heterogeneity on the spatial coverage of such kind of data. So such study cannot be really considered as an independent estimate from previous ones and, should have the same bias, if exists, that other studies.

Response: We agree that the new global R_s estimates based on a random forest algorithm and an updated global daily R_s database cannot be considered as an independent estimate from previous ones, and have revised that paragraph and reinforced that: “but it should be noted that DGRsD is not independent from SRDB, and therefore more evidence is needed to ensure there are no systematic biases in $R_{s,iii}$ ”, please see lines 157-165 in revision as well.

Reviewer: Another argument presented in the case of R_s :GPP ratio explicitly given in the SRDB database or estimated from combination of SRDB with FLUXNET is to explain that there is probably no bias related to spatial extent of data since weighting (or not) the measurements by vegetation areas globally gives similar results. But it just proves that values from underrepresented vegetation type is not very different for those that are overestimated. But if sites on vegetation from which there is only few measurements are not well representative of the whole ecosystem we can still have large biases. We can also notice that if GPP is estimated from the in-situ R_s :GPP ratio of 0.54 and the new global R_s estimate of 93 GtC we find a global GPP value of 172 GtC which is even larger than estimation given from GPP_ $R_{s,lit}$!

Response: This is true—the fact that weighting measurements by vegetation areas globally gives similar results is not dispositive. As the reviewer points out, one can construct scenarios where biases in sampling results in serious biases for global estimates. Nonetheless, we think that our weighting test is useful. We have modified the text to more clearly make this point, please see lines 125-135. In addition, as all CMIP6 models reported R_H :GPP ratio, we therefore also reported R_H :GPP ratio in Figure 2 now, and the new results are described and discussed in the revision (Lines 137-145).

Reviewer: In addition to sources of uncertainties from R_s that are mentioned in the manuscript there is the large spatial heterogeneity of respiration even at a very small spatial scale making chambers measurements not necessarily representative of a stand (in opposite to flux measurement that is more integrative). More important, R_s largely depend on the land use history of the site. This means that R_s :GPP ratio can be largely variable within a given ecosystem and climate condition not because of uncertainty in the measurement but just because of disequilibrium between GPP and heterotrophic respiration. Hence for instance an R_s :GPP ratio on cropland with soil largely depleted in carbon or on a reforested area can be probably around 0.2 whereas, on the opposite, R_s :GPP, in the same conditions, on a recently deforested cropland or on the beginning of a forest rotation can probably reach 0.7.

Response: We agree with the reviewer, site history could have a confounding effect, and it is not consistently reported. However, (1) FLUXNET sites tend to be located in undisturbed (or lightly disturbed) ecosystems (there are exceptions of course, but overall we believe this is a common practice for most sites); (2) an site-history effect would create more noise in the relationship, but wouldn't necessarily induce a bias; (3) usually R_s and GPP are reported from a same study in SRDB, and thus land use and measurement year are the same; and (4) R_s :GPP ratio from SRDB are similar as that from FLUXNET (Figure 2). As in our previous response, we acknowledge that we cannot conclusively prove there is not bias here, but we believe it is quite unlikely. This issue is now discussed in the manuscript (lines 196-215, and 352-357). In addition, other uncertainties are also discussed (Table 2 and Table S3).

Reviewer: So for me uncertainties in estimation of R_s is probably as large as estimation on GPP (even probably higher) and then it is very difficult to conclude from the information given in the paper anything about which one of the terms is the major cause of the discrepancies which limit the main message of the paper. **Probably the best argument is that some recent methods for GPP estimations seems to be more in agreement with estimated GPP deduced from $R_{s\text{lit}}$ (which is in some way contradictory with the title which state that productivity and respiration fluxes are inconsistent)**, I discuss this point more in detail hereafter. But here again these new methods are not necessarily less uncertain than previous one predicting lower global GPP (for instance SIF data is very promising as it is a more direct proxy of GPP than NDVI but still very preliminary and then also very uncertain).

Response: Thanks for these interesting and thoughtful points. We did some additional analysis, weighting the data by both time and GPP methods, and found that the gaps between carbon-cycle flux collected from the literature (GPP_{lit} and $R_{s\text{lit}}$) and the results implied by the other (GPP_{R_s} and $R_{s\text{GPP}}$) did decrease; however, these distributions are still significantly different from each other (see figure below and Figure S9 in the revision). Therefore, we decide not to emphasize

that low GPP is the reason, and instead highlight that either GPP is biased low or R_s is biased high (Table 2).

Reviewer: One important point is that, in opposite to R_s estimation which is only based on direct measurement or estimation for flux tower, GPP can be estimated by different independent methods. This is unfortunately only briefly discussed in the paper and only shown in figure S5. This is however an important point. In particular looking to table S2 we can see that a large majority of estimates used in the study come from MODIS which gives all relatively similar estimations in the lower range of the estimated GPP. Then the mean GPP_{lit} of 113 PgC given as reference calculated by averaging all the GPP estimates from literature is mainly dominated by MODIS estimations and then relatively low. For instance, if instead of averaging all the values we first average the values coming for each method (I.e NDVI, satellite driven models, FLUXCOM, FLUXNET, SIF, O18, IPCC, global carbon) and then averaging the mean value of each method to give a similar weight to each method independently of the number of available estimation the mean value is 130 GtC which is still lower than estimated from R_s but closer however. More generally, as briefly mentioned, some methods like O18 or SIF for instance gives estimates of 162 GtC and 147 GtC which are in agreement with GPP estimated from $R_{s_{lit}}$ of 149GtC ! But as I mentioned previously, it doesn't prove that these values are more realistic than estimation based on remote sensing NDVI which are around 112GtC but at least it shows that the estimations or GPP and R_s are not necessarily fully inconsistent are claimed in the title. Likewise

the fact to say that GPP is probably underestimation is not necessarily true for all the method considered. So it would be important to have a deeper discussion on the different methods of estimation of GPP and not just look to the average of these estimations.

Response: We re-analyzed the data, and found that if GPP_{lit} groups are weighted equally (i.e., aggregated into 6 different groups before bootstrap resampling), the bootstrapped results ($GPP_{lit-group}$) are higher and closer to the GPP_{Rs} (see figure below and Figure S5 in the revision), but this does not change the conclusions of our study—the GPP and R_s literature results remain fundamentally inconsistent with each other. We discuss this in the revision (Lines 167-181 and Table 2).

Reviewer: Concerning the discussion of R_s :GPP simulated by ESMs there is an important limit which is related to the fact that only two versions of the same ESM model (CESM2) is considered in the study. Considering the large spread of modeled respirations in the different ESMs, it is very difficult to give a general conclusion on simulated R_s :GPP ratios in ESMs based only on a single model. Obviously I understand that it is related to the fact that only CESM2 report the partition of respiration within the different biomass compartments in the CMIP6 database. However there is some ways to see how CESM2 behave (in term of ratio between

respiration and GPP) compared to others models. **For instance, one can see how ratio of total Ra to GPP and Rh to GPP in CESM2 compare to others models.** Probably, most of the models provide also the partition of biomass between the different compartments. Even if respiration coefficient can be different for the different compartments, weighting Ra with the relative biomass of each compartment could be a first guess of root respiration as well. So to be able to discuss what is the ratio of Rs to GPP from the models point of view it is very important that CESM2 could be compared to others ESMs to see if it is, for the different ratio of respiration to GPP, in the middle of models range or at one of the end members. In particular from what I understand in the text Ra_root:GPP is 21% and Rs:GPP 27-29% that would means that Rh:GPP is only 6-8 % which seems unrealistic.

Response: In the revision, we also analyzed the ratio between R_H and GPP (bottom panel of Figure 2 in the revision), and we have now a total of 104 model \times ensemble combinations (16 models). The R_H :GPP from CMIP6 models (both local and global scale) are similar to those from the literature (in contrast to the Rs:GPP ratios, which are lower in models), but higher than those from SRDB and FLUXNET reported R_H :GPP ratio. We discussed this in the revision (lines 116 – 145).

Reviewer: About site-level data it is mentioned that FLUXNET GPP was linked to an SRDB Rs if both measurements occurred within 5km in the same vegetation type and the same year. However, as I mentioned previously, there is another criteria that should be also taken into account which is the land use history of the site. Indeed if a land conversion occurred in the last decades, Rs will not be in equilibrium with GPP making the Rs:GPP ratio incorrect. This is obviously a difficult question as the site history is not necessarily reported. The fact that the estimated Rs:GPP ratio estimated is similar to value reported in the SRDB data (for which I guess both Rs and GPP are reported from the same site) seems to indicate that this not induce an important bias but this point should be considered carefully and discussed however.

Response: We believe it is unlike the land use history causes an important bias because 1) usually R_s and GPP are reported from a same study in SRDB, and thus land use and measurement year are the same; and 2) R_s :GPP ratio from SRDB are similar as that from FLUXNET (Figure 2). This issue is now explained in the manuscript (lines 352-357).

Reviewer: In summary the paper make an interesting review of our existing knowledge on both global GPP and soil respiration. It shows in particular that there is still very large uncertainties in the different terms making Rs and GPP difficult to reconcile at global scale. However there is some assumptions or conclusions that are not very convincing especially the fact that the most likely “suspect” would be a too low estimate of GPP. Soil respiration measurement it, at least, as uncertain as GPP. Moreover, since there is several methods to estimate GPP, we can have an estimation of the possible range which is not the case for respiration for which only extrapolation from in-situ measurement are available and then few means to access its effective uncertainty. Some methodological aspect should also be revised. **This is the case of evaluation of Rs:GPP ratio from ESM that should not only based on a single model. It is also important to discuss ratio of Rs:GPP in regard of the different methods for estimation of GPP.**

Response: Thanks for the insightful comments, which are extremely useful to improve the quality of the manuscript. We have added some additional analysis and strengthened the

discussion accordingly. Briefly: 1) R_S from CMIP6 models was calculated based on R_H and $R_{root}:R_S$ ratio using a bootstrap approach, and now we have much more $R_S:GPP$ ratio data from CMIP6 models ($n=40352$, shown in Figure 2) (i.e., we resampled a $R_{root}:R_S$ ratio from data shown in Figure S2c to calculate R_S of each CMIP6 model outputs); 2) as all CMIP6 models reported R_H values, we therefore plotted $R_H:GPP$ ratio in Figure 2 (bottom panels). In the revision, the new results are talked in details (Lines 116-145). Please also see our response to each comment above.

Reviewer #3 (Remarks to the Author):

Reviewer: Summary:

The authors present a compilation of literature values for GPP and soil respiration (R_S) and show that they are not consistent with one-another. They find a mismatch; that either GPP is too small or R_S is too large. They conclude that the most likely scenario is that GPP is currently underestimated when considering potential biases of both global GPP and R_S estimates. They also show that ratios of R_S/GPP are highly variable across methods: 0.54 for site-level observations, 0.71 for global estimates, and 0.28 for modeled estimates.

Response: We thank the third reviewer for the insightful comments on our manuscript.

Reviewer: General strengths:

The approach is novel, and to my knowledge, has not been done before. The compilation effort is nice. The GPP and R_S measurement communities are operating pretty independent of one another so this comparison is extremely useful. The range of inconsistencies that the authors point out is troubling for the field.

Response: Thanks for the overall positive opinion on this work.

Reviewer: General comments:

I previously reviewed this manuscript when the authors submitted to a different journal. I've compared my previous comments to this revised version and most of my questions have been answered in the new submission, therefore this review is unusually brief. They have added a discussion of why the R_S and GPP estimates may not reconcile which adds value to the paper in my opinion. I think this would make a nice contribution to Nature Communications in its current form.

Response: Thanks for taking the time to review our manuscript again. We appreciate the constructive comments you gave in the first round of review process, and really appreciated your affirmation of our revision.

Reviewer: Specific comments:

Reviewer: Line 137: ...implied by atmospheric $18O:16O$ ratios of CO_2 ...

Response: Resolved (Line 151).

Reviewer: The figure captions are a little hard to understand and could use a grammar check.

Response: We have checked all figures' caption for clarity, please see the tracked change version of the manuscript for details.

Reviewer: Figure 2: global CMIP6 results do not appear to be dark blue like the caption indicates. Perhaps the symbols are simply behind the grey box-and-whisker plot though.

Response: That's right, we changed "dark blue" to "blue" in the caption.

Reviewer: Table 2: I don't find this table especially interesting to warrant including in the main text. It would be fine in supplemental.

Response: We changed our argument and highlight that either GPP is biased low or R_S is biased high, therefore, we redesigned Table 2, only those possibilities support above argument are kept, we now use this Table 2 summarized the core message of this manuscript.

Reviewer: Fig S5: Are atmospheric constraint estimates missing from this figure? What is FLUXNET isotope data?

Response: In the previous version, atmospheric constraint estimates are grouped into Site-based upscaling. "FLUXNET isotope data" should be "FLUXNET and isotope data". GPP from literature are regrouped into 6 categories now in the revision (Figure S5).

Reviewers' Comments:

Reviewer #2:

Remarks to the Author:

The revised manuscript greatly improved and correctly replied to issues raised on my first review process. Especially:

1/ the new analysis of bootstrapping results considering relative weight of different GPP methods estimates (as well as another important point raised by reviewer 1 relative to the difference in the period covered by GPP and Rs) make the paper results and discussion definitely more robust! Taking into account for these factors, the reconstructed GPP and Rs are not so different even it doesn't change the main conclusion of the paper about the discrepancies between both. One can also notice that it reduces the dispersion of the results. These results are presented in the discussion but, as we could guess that this new estimation is probably more realistic than the original one, I don't know if it would be better to present these results in figure 1 (and give these numbers in the abstract) and then put the original results in the discussion ?

I have just a small question about the reconstruction with aggregated years. Like pointed by reviewer 1 I would expect that, taking into account the difference in Rs and GPP period, tend to increase Rslit and GPPrs and on the opposite decrease GPplit and increase GPPrs. So the results are counterintuitive. On the other hand the bootstrapping method is complex and then probably results can not just be deduced in a such easy guess from expected GPP and RS trends... But I would be, by curiosity interested to understand why ?

Concerning the comparison between GPP estimation methods. In addition to showing the discrepancies between estimation of GPP and Rs, an important outcome of the paper is that it gives a constraint on validity of the different GPP estimations. Indeed we clearly see that MODIS estimations largely underestimate GPP where SBU and SIF estimation seems to be more realistic as they fit better with reconstructed GPP from RS. So I think this important outcome could be better highlighted in the paper.

2/ The analysis from CMIP6 models is now very interesting and very much robust ! In particular it is reassuring to see that the model ensemble gives a Rs to GPP which is in agreement with different types of data (it is true that it is from a statistical point of view lower than SRBD and fluxnet, but very close however in contrary to Rslit/gplit ratio !, so this could be mentioned in the text). As there is now a comprehensive estimation of GPP and Rs from models, I think that it would be an added value to put these global estimations of GPP and Rs in figure 1 ! Then we could have a comprehensive view of the different ways to estimate GPP and RS by observations and models.

Another very new interesting outcome is the analysis of Rh to GPP ratios showing, in contrary to Rs:GPP, that Rh:GPP ratios from models are clearly overestimated and, as mentioned in the text, indicates a too low Ra from roots. This is a very important result for modellers as this gives a new constraint for models parameterization!

In summary there are several new important results and the different analyses are more robust. I think that some important results could be better highlighted and the Figure 1 would benefit of including also CMIP6 model results but except for these minor things I find that the revised paper is now in good shape for publication.

Reviewer #3:

Remarks to the Author:

Review of Jian et al. revision 1 to Nature Communications.

Summary:

I (Reviewer #3) was asked by the editor to respond to Reviewer #1's concerns. The main ones were: 1) bias introduced by temporal trends in global C fluxes, and 2) partitioning ratio uncertainty. The authors did attempt to address these points, but I find that the additional analysis and discussion did

not clearly resolve these issues. I've added 2 new comments: 3) bias in Rs observations based on my recent literature findings. And 4) provided a brief comment of my own regarding the issue of the misleading title mentioned by Reviewer #2.

1) Bias introduced by temporal trends in global C fluxes:

Reviewer #1 brings up an important concern regarding the temporal trends in GPP and Rs that have been occurring as a result of drivers like increased global temperatures, atmospheric CO₂ concentrations, and N fertilization (e.g. Campbell et al. 2017). The observed increase in the global land sink constrains the trends such that the trend in GPP is greater than the trend in Re (Rh+Ra). There are few details about how the authors created Fig S8 above. My impression is that they aggregated the estimates by year (or created a binned average by year) and then resampled the aggregated lit estimates in the similar manner that did in the original analysis. Unfortunately, this was the wrong approach to take. This still samples Rs more often in the 1990s and samples GPP more often in the early 2000s. To appropriately address Reviewer #1's concern, the authors should have normalized the lit estimates to the same year(s). In other words, they should attempt to estimate true GPP and Rs fluxes during the same decade. The confounding factor here is that the approaches for estimation have varied over time as well. Therefore, the trends the authors plot in Fig S8 are a combination of true increases in fluxes overlaid with different estimation approaches which have their own temporal trends. There is no easy way to account for these issues in the lit values. However, the direction of the correction that the authors' find (i.e. closing the gap between GPP and Rs lit values) is the opposite of what we would expect given the 'true' trends in GPP and Re. At the very least, these decadal trends in GPP and Re and their impact on the results of this study (to widen the gap, contrary to the analysis presented in Fig S8) should be discussed.

Lines 209-215 and Fig S8 should be revised.

Campbell, J E, J A Berry, U Seibt, S J Smith, S A Montzka, T Launois, S Belviso, L Bopp, and M Laine. "Large Historical Growth in Global Terrestrial Gross Primary Production." *Nature* 544, no. 7648 (2017): 84-87. <https://doi.org/10.1038/nature22030>.

2) Partitioning ratio uncertainty:

The authors show that there is a lack of partitioning ratios at low GPP sites. However, the biggest influence is likely the enormous ranges of partitioning ratios even where there is data (from nearly zero to nearly 1). Showing a trend in Rroot:Ra is a distraction from the main uncertainty discussion that needs to be present, in my opinion.

3) New consideration of Rs uncertainty:

I've recently become aware of abiotic processes that contribute to a disconnect between the CO₂ produced in the soil by root and heterotrophic respiration and that measured as a soil CO₂ flux out of the surface (e.g. calcite dissolution in the soil). Please see Sánchez-Cañete et al. (2018) for a discussion of this bias. It would result in a bias in the Rs databases being too low. On one hand, this seems even harder to close the budget. But on the other, if it resulted in higher estimates of Rs:GPP ratios for site level obs, would that reconcile the lit values of Rs:GPP? I'm not sure if that's helpful, but it seems like a directional bias that should be mentioned.

Sánchez-Cañete, Enrique P., Greg A. Barron-Gafford, and Jon Chorover. "A Considerable Fraction of Soil-Respired CO₂ Is Not Emitted Directly to the Atmosphere." *Scientific Reports* 8, no. 1 (December 2018): 13518. <https://doi.org/10.1038/s41598-018-29803-x>.

4) Comment on the title and main message

Reviewer #2 states "Probably the best argument is that some recent methods for GPP estimations seems to be more in agreement with estimated GPP deduced from Rslit (which is in some way contradictory with the title which state that productivity and respiration fluxes are inconsistent)." I think this is a fair point and it could be addressed in 2 ways. First, the title could include a word like 'central estimates' or similar (not sure that's the best word) to denote that the authors are weighting the entire body of literature from the 1980s to present equally. And, edit paragraph lines 167-181 to discuss "more newly published" estimates like they do for Rs in the following paragraph.

Minor comments:

Table 2: The discussion points are a combination of biases but also just general large uncertainty in some of these estimates. The title should include large sources of uncertainty in addition to directional bias. As stated above, the temporal trends in Rs and GPP were misinterpreted by the authors, therefore the first point in Rs too high and the last point in GPP too low.

Fig 2: Is there a brief explanation of the 3 cluster groupings in the CMIP6 model estimates in the lower panel? Are those 3 different model versions?

Typos: line 188 (nevertheless), line 139 (either)

REVIEWER COMMENTS

Reviewer #2 (Remarks to the Author):

The revised manuscript greatly improved and correctly replied to issues raised on my first review process. Especially:

1/ the new analysis of bootstrapping results considering relative weight of different GPP methods estimates (as well as another important point raised by reviewer 1 relative to the difference in the period covered by GPP and Rs) make the paper results and discussion definitely more robust! Taking into account for these factors, the reconstructed GPP and Rs are not so different even it doesn't change the main conclusion of the paper about the discrepancies between both. One can also notice that it reduces the dispersion of the results.

Response: We agree, and appreciate the enthusiasm about this improvement.

These results are presented in the discussion but, as we could guess that this new estimation is probably more realistic than the original one, I don't know if it would be better to present these results in figure 1 (and give these numbers in the abstract) and then put the original results in the discussion?

Response: This is possible, but after consideration we decided not to. Like Reviewer 2 (cf. following comment and response) as well as the new Reviewer 3, we recognize that the resampling produces counterintuitive results, as well as adding another layer of numerical processing and thus, to some degree, uncertainty. For these reasons we prefer to retain the 'raw' data analysis as our primary one, while making extensive reference in the discussion to the two alternative resampling analyses (controlling for methods and temporal periods, see lines 211-223) to provide readers with this crucial context.

I have just a small question about the reconstruction with aggregated years. Like pointed by reviewer 1 I would expect that, taking into account the difference in Rs and GPP period, tend to increase R_{slit} and GPP_{rs} and on the opposite decrease GPP_{lit} and increase GPP_{rs} . So the results are counterintuitive. On the other hand the bootstrapping method is complex and then probably results can not just be deduced in a such easy guess from expected GPP and RS trends... But I would be, by curiosity interested to understand why?

Response: We agree that the results are counterintuitive but they may be useful for discussion and exploration in future research. That said, we did some additional analysis to verify our results. As suggested by reviewer #3, we now standardized R_s and GPP to the same years (i.e., grouped to 1986, 1989, 1992, 1995, 2001, 2007, 2010, and 2013, as showed in Figure 1 below), but we still get the same outputs (narrow down the gap, rather than widen the gap). The possible reason is that the mean value of the raw GPP values (same logic for Rs data) collected from literature is 120 Pg C yr^{-1} , while the mean value of the GPP after aggregation is 127 Pg C yr^{-1} (see the density distribution in Figure 1). Bootstrapping is an approach to resampling and estimate the mean of the data, as the mean of GPP after aggregation (127 Pg C yr^{-1}) is bigger than the raw GPP (120 Pg C yr^{-1}), therefore, taking temporal adjustment decrease the gap rather than increase the gap.

As noted above, the bootstrapping step is complex and diminishes the analytical clarity (while bringing undeniable benefits; c.f. Efron 1985 https://link.springer.com/article/10.2333/bhmk.12.17_1 and many subsequent papers). For this reason we have adopted Reviewer 3's suggestion of noting that, as Reviewer 2 points out as well, any temporal adjustment will almost increase—not decrease—the significance of our results, while presenting the 'raw' unadjusted analysis as our primary result. We believe this strikes a reasonable balance between clarity and robustness when, as Reviewer 3 notes, a full correction for changing methodologies, carbon cycle, etc., over time is probably not possible.

Figure 1. Comparison between raw GPP data collected from literature vs aggregated GPP by year.

Concerning the comparison between GPP estimation methods. In addition to showing the discrepancies between estimation of GPP and R_s , an important outcome of the paper is that it gives a constraint on validity of the different GPP estimations. Indeed we clearly see that MODIS estimations largely underestimate GPP where SBU and SIF estimation seems to be more realistic as they fit better with reconstructed GPP from RS. So I think this important outcome could be better highlighted in the paper.

Response: This is a good point, and we now do so in lines 170, and 242-248.

2/ The analysis from CMIP6 models is now very interesting and very much robust ! In particular it is reassuring to see that the model ensemble gives a R_s to GPP which is in agreement with different types of data (it is true that it is from a statistical point of view lower than SRBD and fluxnet, but very close however in contrary to R_{slit}/gpp_{lit} ratio !, so this could be mentioned in the text). As there is now a comprehensive estimation of GPP and R_s from models, I think that it would be an added value to put these global estimations of GPP and R_s in figure 1 ! Then we could have a comprehensive view of the different ways to estimate GPP and R_s by observations and models.

Response: This is a great point! However, here we only extracted CMIP6 data for the SRDB and FLUXNET sites—that is to say, we did not use the models' entire data output. In addition, only a small number of models provide R_s (it is much more common to provide only R_h). Therefore, we decided not plot out the GPP and R_s of CMIP6 models in Figure 1. We now clarified this point in the methods; please see lines 323-325.

Another very new interesting outcome is the analysis of R_h to GPP ratios showing, in contrary to R_s :GPP, that R_h :GPP ratios from models are clearly overestimated and, as mentioned in the text, indicates a too low R_a from roots. This is a very important results for modeller as this give a new constraint for models parameterization!

Response: Thanks.

In summary there is several new important results and the different analysis are more robust. I think that some important results could be better highlighted and the Figure 1 would benefit of including also CMIP6 model results but except for these minor things I find that revised paper is now in good shape for publication.

Response: We appreciate for the overall positive opinion on this revision from reviewer 2.

Reviewer #3 (Remarks to the Author):

Review of Jian et al. revision 1 to Nature Communications.

Summary:

I (Reviewer #3) was asked by the editor to respond to Reviewer #1's concerns. The main ones were: 1) bias introduced by temporal trends in global C fluxes, and 2) partitioning ratio uncertainty. The authors did attempt to address these points, but I find that the additional analysis and discussion did not clearly resolve these issues. I've added 2 new comments: 3) bias in Rs observations based on my recent literature findings. And 4) provided a brief comment of my own regarding the issue of the misleading title mentioned by Reviewer #2.

1) Bias introduced by temporal trends in global C fluxes:

Reviewer #1 brings up an important concern regarding the temporal trends in GPP and Rs that have been occurring as a result of drivers like increased global temperatures, atmospheric CO₂ concentrations, and N fertilization (e.g. Campbell et al. 2017). The observed increase in the global land sink constrains the trends such that the trend in GPP is greater than the trend in Re (Rh+Ra). There are few details about how the authors created Fig S8 above. My impression is that they aggregated the estimates by year (or created a binned average by year) and then resampled the aggregated lit estimates in the similar manner that did in the original analysis. Unfortunately, this was the wrong approach to take. This still samples Rs more often in the 1990s and samples GPP more often in the early 2000s. To appropriately address Reviewer #1's concern, the authors should have normalized the lit estimates to the same year(s). In other words, they should attempt to estimate true GPP and Rs fluxes during the same decade. The confounding factor here is that the approaches for estimation have varied over time as well. Therefore, the trends the authors plot in Fig S8 are a combination of true increases in fluxes overlaid with different estimation approaches which have their own temporal trends. There is no easy way to account for these issues in the lit values. However, the direction of the correction that the authors' find (i.e. closing the gap between GPP and Rs lit values) is the opposite of what we would expect given the 'true' trends in GPP and Re. At the very least, these decadal trends in GPP and Re and their impact on the results of this study (to widen the gap, contrary to the analysis presented in Fig S8) should be discussed.

Lines 209-215 and Fig S8 should be revised.

Campbell, J E, J A Berry, U Seibt, S J Smith, S A Montzka, T Launois, S Belviso, L Bopp, and M Laine. "Large Historical Growth in Global Terrestrial Gross Primary Production." Nature 544, no. 7648 (2017): 84–87. <https://doi.org/10.1038/nature22030>.

Response: This is a good point, one also made by Reviewer 2, and we agree with it. To summarize:

- It's possible that the temporal discrepancies in the GPP and Rs data could confound the analysis; but
- There is no easy way to fully control for the varying sampling rates and methods over time; clearly, however,

- Conceptually, correcting for the temporal issue should increase, not decrease, the significance (magnitude) of our results.

In summary, then, and as we respond to Referee 2 above, we think a reasonable path forward is to present the ‘raw’ unadjusted analysis as our primary result, while noting that any temporal adjustment will almost *increase*—not decrease—the numerical significant differences of our results (Figure 1 and Figure 2 in this document), and referencing the supplementary re-analyses we performed based on temporal and methodological factors. We believe this strikes a reasonable balance between clarity and robustness when, as you note, a full correction for changing methodologies, carbon cycle differences, etc., over time is probably not possible. We discuss this in lines 211-223.

Figure 2. Global annual soil respiration (R_s) and gross primary productivity (GPP) collected from the literature and its trend over time (between 1980 and 2017). The analysis of temporal representativeness affects bootstrap resampling results of GPP and R_s from the literature (GPP_{lit} and $R_{s\text{lit}}$). GPP_{lit} and $R_{s\text{lit}}$: bootstrap resampling results of GPP and R_s based on the raw

data; $GPP_{lit-agg}$ and $R_{s,lit-agg}$: aggregate R_s and GPP by year first, then using bootstrap resampling to get the results for GPP and R_s .

2) Partitioning ratio uncertainty:

The authors show that there is a lack of partitioning ratios at low GPP sites. However, the biggest influence is likely the enormous ranges of partitioning ratios even where there is data (from nearly zero to nearly 1). Showing a trend in $R_{root}:R_a$ is a distraction from the main uncertainty discussion that needs to be present, in my opinion.

Response: We agree and now we only focus on the representativeness of $R_{root}:R_s$, $R_{root}:R_a$, and $R_a:GPP$ but removed the linear regression analysis between GPP and those ratios, please see lines 225-236 in the revision.

3) New consideration of R_s uncertainty:

I've recently become aware of abiotic processes that contribute to a disconnect between the CO_2 produced in the soil by root and heterotrophic respiration and that measured as a soil CO_2 flux out of the surface (e.g. calcite dissolution in the soil). Please see Sánchez-Cañete et al. (2018) for a discussion of this bias. It would result in a bias in the R_s databases being too low. On one hand, this seems even harder to close the budget. But on the other, if it resulted in higher estimates of $R_s:GPP$ ratios for site level obs, would that reconcile the lit values of $R_s:GPP$? I'm not sure if that's helpful, but it seems like a directional bias that should be mentioned.

Sánchez-Cañete, Enrique P., Greg A. Barron-Gafford, and Jon Chorover. "A Considerable Fraction of Soil-Respired CO_2 Is Not Emitted Directly to the Atmosphere." *Scientific Reports* 8, no. 1 (December 2018): 13518. <https://doi.org/10.1038/s41598-018-29803-x>.

Response: We appreciate the reviewer raising this possibility. This is an interesting study, but we must note that this is a site-specific study performed in a water-limited ecosystem. Although abiotic processes are important for the net soil CO_2 efflux in several ecosystems, it has not been proven that *two-thirds* of R_s is not being measured correctly globally. We respectfully disagree that this particular study demonstrates that *two-thirds* of R_s is not being measured correctly globally and for this reason, while we do now cite it (ref 37 in line 213 and Table S3) as one of many potential methodological uncertainties, we do not place too much weight on this potential challenge.

4) Comment on the title and main message

Reviewer #2 states "Probably the best argument is that some recent methods for GPP estimations seems to be more in agreement with estimated GPP deduced from R_{slit} (which is in some way contradictory with the title which state that productivity and respiration fluxes are inconsistent)." I think this is a fair point and it could be addressed in 2 ways. First, the title could include a word like 'central estimates' or similar (not sure that's the best word) to denote that the authors are weighting the entire body of literature from the 1980s to present equally. And, edit paragraph lines 167-181 to discuss "more newly published" estimates like they do for R_s in the following paragraph.

Response: We have modified the title (“Historically inconsistent”), and have now emphasize this point—which we agree is a fair one—in a number of places: see lines 4, 41, 44, 149, and 309.

Minor comments:

Table 2: The discussion points are a combination of biases but also just general large uncertainty in some of these estimates. The title should include large sources of uncertainty in addition to directional bias. As stated above, the temporal trends in Rs and GPP were misinterpreted by the authors, therefore the first point in Rs too high and the last point in GPP too low.

Response: We have clarified and expanded the table caption to address this point, and removed entirely the points about temporal discrepancies.

Fig 2: Is there a brief explanation of the 3 cluster groupings in the CMIP6 model estimates in the lower panel? Are those 3 different model versions?

Response: We believe this is basically correct; Earth System Models tend to cluster into a smaller number of ‘families’ produced by different modeling centers with similar behavioral characteristics (see for example Lehner et al. 2020 <http://dx.doi.org/10.5194/esd-11-491-2020> and Todd-Brown et al. 2013 <https://bg.copernicus.org/articles/10/1717/2013/>), and the distribution on this plot reflects that.

Typos: line 188 (nevertheless), line 139 (either)

Response: fixed, thanks.

Reviewers' Comments:

Reviewer #2:

Remarks to the Author:

After the first paper revision, the manuscript was largely improved compared to the original one. I had then only few comments from which the authors replied in an acceptable way and the new improvements of the manuscript is fine for me. So I have no more comments on the new version of the manuscript and agree with its publication in the present form.

Reviewer #3:

Remarks to the Author:

The authors improved the aggregated bootstrap analysis and the text revisions are a fair summary of the results. For the record, I don't think the aggregated bootstrap analysis addresses the issue of underlying trends in GPP or R_s over time. However, I think the readers will be able to see that higher GPP estimates, either by new methods yielding higher values (and weighting those more in the aggregate analysis), or an increase in GPP over time due to climate response would tend to close the gap with R_s estimates.

I think the manuscript is ready to go, except for a few minor suggestions for the authors to consider.

Line 109-116: Are the Root/RA variance or uncertainty really tied to specific ecosystems like 'other = cropland, desert, wetland, and savanna' as indicated in Table 1? If so, consider highlighting these specific ecosystems that are perhaps understudied. But note that it is likely the ones with largest C fluxes that likely have the biggest lever here, e.g. croplands.

Line 113: Do you mean the variance in R_{slit} estimates or the uncertainty (i.e. spread of estimates)?

Line 115: Root/RA is listed in Table 1, not Root/ R_s as in the text

Line 132: text says 'local/grid scale', Fig 2 says 'ecosystem scale'. Choose consistent terminology.

Line 140-141: "Too low RA values" implies that lit R_s /GPP is correct and CMIP models are wrong. But the models agree with SRDB and fluxnet. Is this the intent? Consider rephrasing.

Line 184: citation needed at the end of the sentence? Table S3? Table S3 could include the saturation of NDVI signal for example.

Lines 211-223: As written, the increase in GPP estimates over time is attributed to new/different methods. What is not mentioned in that GPP could be increasing over time (see previous comment and citation). If there is no documented trend in R_s (citation?) and if GPP is increasing over time due to temp increase or CO₂ fertilization, etc., then the R_s -GPP gap could be closing.

Lines 226-229: In the text, the following are listed as significant uncertainties: Root/Re, Root/ R_s , and RA/GPP. These are different from what appears in Table 1 and Fig 2: Root/RA, Root/ R_s , NPP, and lit estimates. Please make these consistent.

Lines 225-236: The focus on low productivity regions here is confusing. It's likely that regions with significant C fluxes will be more influential. Table 2 identifies other/crops + grasslands. It would make more sense to comment on the 'other' than low productivity regions.

Dear Dr. Frischkorn, thank you for your assessment. We believe we have comprehensively addressed the reviewers' concerns. Please see our responses to the comments below.

Reviewer #2 (Remarks to the Author):

After the first paper revision, the manuscript was largely improved compared to the original one. I had then only few comments from which the authors replied in an acceptable way and the new improvements of the manuscript is fine for me. So I have no more comments on the new version of the manuscript and agree with its publication in the present form.

Response: We appreciate reviewer#2 for his positive opinion on our revision.

Reviewer #3 (Remarks to the Author):

The authors improved the aggregated bootstrap analysis and the text revisions are a fair summary of the results. For the record, I don't think the aggregated bootstrap analysis addresses the issue of underlying trends in GPP or R_s over time. However, I think the readers will be able to see that higher GPP estimates, either by new methods yielding higher values (and weighting those more in the aggregate analysis), or an increase in GPP over time due to climate response would tend to close the gap with R_s estimates.

I think the manuscript is ready to go, except for a few minor suggestions for the authors to consider.

Response: We appreciate reviewer#3 for their time and helpful comments in reviewing our manuscript. We agree that readers will be able to see that current GPP estimates from literature are too low to close the gap between GPP and R_s . However, in this manuscript we are not able to confirm this argument, so we do not conclude that currently GPP estimates are too low.

Line 109-116: Are the R_{root}/R_A variance or uncertainty really tied to specific ecosystems like 'other = cropland, desert, wetland, and savanna' as indicated in Table 1? If so, consider highlighting these specific ecosystems that are perhaps understudied. But note that it is likely the ones with largest C fluxes that likely have the biggest lever here, e.g. croplands.

Response: Yes, $R_{root}:R_A$ (other) including cropland, savanna, grassland, and wetland, as shown in Figure S3. This section has been revised as: "Other influential variables were variance in global $R_{S_{lit}}$ (12%) and the root contribution to total R_S of desert, wetland, and savanna (7%). For bootstrapped $R_{S_{GPP}}$, uncertainty in GPP_{lit} was the largest (35%) contributor to variability, with root contribution to total R_A of cropland, savanna, grassland, and wetland (32%) and global NPP (28%) also large. No other factor contributed more than 2% for variability in GPP_{R_S} ." (Lines 118-122).

We agree that it is a good idea to highlight ecosystems that are perhaps understudied. In the revision, we highlighted this point: "inferring GPP from $R_{S_{lit}}$ and inferring R_S from GPP_{lit} (Table 1) also show that $R_{root}:R_S$ and $R_{root}:R_A$ measurements from desert, wetland, cropland, and savanna are key variables to close the gap between productivity and respiration fluxes in the global terrestrial carbon cycle. In addition, arctic regions and Tibetan Plateau store large amount of organic matter and are experiencing fast climate change. In the future, increasing field measurements of $R_{root}:R_S$, $R_{root}:R_A$, and $R_A:GPP$, especially in low-productivity regions, arctic

regions, and Tibetan Plateau is important to close the terrestrial carbon budget”, please see lines 247-253 in the revision.

Line 113: Do you mean the variance in Rslit estimates or the uncertainty (i.e. spread of estimates)?

Response: This should be variance. We have changed “uncertainties” to “variance” (Line 118).

Line 115: Rroot/RA is listed in Table 1, not Rroot/Rs as in the text

Response: This is an error. We have corrected it (line 120).

Line 132: text says ‘local/grid scale’, Fig 2 says ‘ecosystem scale’. Choose consistent terminology.

Response: “grid cell site-level” was used consistently in the manuscript now.

Line 140-141: “Too low RA values” implies that lit Rs/GPP is correct and CMIP models are wrong. But the models agree with SRDB and fluxnet. Is this the intent? Consider rephrasing.

Response: Here we mean that from Figure 2d, we can see R_H :GPP of CMIP6 is not significantly different from R_H :GPP of Global. However, from Figure 2b (Lines 145-149), it is clear that R_S :GPP of CMIP6 is lower than R_S :GPP of Global, as $R_S = R_A(\text{from root}) + R_H$. We therefore infer that $R_A(\text{root})$ from CMIP6 maybe too low. We have revised this paragraph, with Figure 2b and Figure 2d marked. We also changed R_A to R_{root} to improve clarity (see lines 147).

Line 184: citation needed at the end of the sentence? Table S3? Table S3 could include the saturation of NDVI signal for example.

Response: Table 2 and Table S3 have been cited at the end of the sentence (Line 192).

Lines 211-223: As written, the increase in GPP estimates over time is attributed to new/different methods. What is not mentioned in that GPP could be increasing over time (see previous comment and citation). If there is no documented trend in Rs (citation?) and if GPP is increasing over time due to temp increase or CO2 fertilization, etc., then the Rs-GPP gap could be closing.

Response: Theoretically, if R_S showed no trend and GPP showed an increasing trend, the R_S -GPP gap could be closing. However, most studies showed that global R_S has an increasing trend under global climate change (see references below).

Bond-Lamberty, Ben, and Allison Thomson. "Temperature-associated increases in the global soil respiration record." *Nature* 464, no. 7288 (2010): 579-582.

Bond-Lamberty, Ben, Vanessa L. Bailey, Min Chen, Christopher M. Gough, and Rodrigo Vargas. "Globally rising soil heterotrophic respiration over recent decades." *Nature* 560, no. 7716 (2018): 80-83.

Hashimoto, Shoji, Nuno Carvalhais, Akihiko Ito, Mirco Migliavacca, Kazuya Nishina, and Markus Reichstein. "Global spatiotemporal distribution of soil respiration modeled using a global database." *Biogeosciences* 12, no. 13 (2015): 4121-4132.

Jian, Jinshi, Meredith K. Steele, Susan D. Day, and R. Quinn Thomas. "Future global soil respiration rates will swell despite regional decreases in temperature sensitivity caused by rising temperature." *Earth's Future* 6, no. 11 (2018): 1539-1554.

Huang, Ni, Li Wang, Xiao-Peng Song, T. Andrew Black, Rachhpal S. Jassal, Ranga B. Myneni, Chaoyang Wu et al. "Spatial and temporal variations in global soil respiration and their relationships with climate and land cover." *Science advances* 6, no. 41 (2020): eabb8508.

Lines 226-229: In the text, the following are listed as significant uncertainties: Root/Re, Root/Rs, and RA/GPP. These are different from what appears in Table 1 and Fig 2: Root/RA, Root/Rs, NPP, and lit estimates. Please make these consistent.

Response: This is a good point. We have changed "ecosystem respiration" to "autotrophic respiration" (Line 237). Now, it is consistent with Table 1 and Figure S7.

Lines 225-236: The focus on low productivity regions here is confusing. It's likely that regions with significant C fluxes will be more influential. Table 2 identifies other/crops + grasslands. It would make more sense to comment on the 'other' than low productivity regions.

Response: Here, we want to point out that relatively few soil respiration measurements are available in the low production area (such as arid region), arctic regions, and Tibetan Plateau, and recent studies showed that this uneven spatial distribution of data likely introduces uncertainty in global soil respiration estimates. We agree this point is not clearly explained in the manuscript, and we have now revised the statement to read: "Recent studies have showed that R_S are relatively less measured in low-productivity regions, arctic regions, and Tibetan Plateau, and that this uneven spatial distribution of data may create large uncertainties when scaling up and estimating global R_S ^{33, 40}, inferring GPP from $R_{S_{lit}}$ and inferring R_S from GPP_{lit} (Table 1) also show that $R_{root}:R_S$ and $R_{root}:R_A$ measurements from desert, wetland, cropland, and savanna are key variables to close the gap between productivity and respiration fluxes in the global terrestrial carbon cycle. In addition, arctic regions and Tibetan Plateau store large amount of organic matter and are experiencing fast climate change. In the future, increasing field measurements of $R_{root}:R_S$, $R_{root}:R_A$, and $R_A:GPP$, especially in low-productivity regions, arctic regions, and Tibetan Plateau is important to close the terrestrial carbon budget". (see lines 244-253).